# Refining Heuristic-Based Bitcoin Address Clustering with Graph Neural Networks

## Abstract

Bitcoin's pseudonymous nature makes it challenging to analyze user-level activity, since a single user may control multiple identifiers (addresses). Existing heuristic-based methods attempt to identify addresses belonging to the same user, but they often produce flat cluster assignments with limited modularity and are prone to errors such as merging different users together. In this work, we propose a method for refining heuristic-obtained clusters by grounding our clustering on contrastive embeddings yielded by graph neural networks. Our contribution is threefold: (i) we release a publicly available dataset of Bitcoin transaction graphs containing a substantial number of clusters; (ii) we propose a methodology for learning address embeddings consistent with heuristics, and back it up with theoretical guiding intuitions; (iii) through hierarchical clustering, we allow a finer analysis of heuristic clusters and provide a quantitative criterion for flagging suspicious merges.

## 1 Introduction

Bitcoin (Nakamoto, 2009) is the first and most widely adopted cryptocurrency, designed as a decentralized payment system without reliance on a central authority. Its operation is enabled by a peer-to-peer network that collectively maintains a shared, immutable record of transactions (Antonopoulos, 2017a). This record, known as the blockchain, provides transparency and auditability while preserving a certain level of pseudonymity for its users; it is organized as a chronological sequence of blocks, each batching the transactions that happened during a certain time interval.

**Bitcoin Address Clustering.** Bitcoin transactions are pseudonymous in nature, as users are identified by random pseudonyms called addresses (Antonopoulos, 2017b). A single user can reuse an address or generate new ones at any time; it is therefore common for a user to control many different addresses. Since addresses are generated randomly, there is no direct way to associate multiple addresses with the same user. While analyzing transaction at the address level can be informative, a user-level analysis provides greater insights. The task of *addresses clustering* consists in grouping together addresses that belong to the same user (without necessarily identifying said user).

**Graph Construction from Transactions.** Graph-based representations are particularly well suited for visualizing and analyzing blockchain data. Two primary types of graphs are commonly employed: those where nodes represent transactions and edges represent the moving bitcoin amounts (Weber et al., 2019), and those where nodes represent users and edges represent transactions (Bellei et al., 2024; Schnoering & Vazirgiannis, 2025). In this paper, we focus on the latter, as it offers a more intuitive representation. Constructing a user-level graph from a set of transactions $\mathcal{T}$ typically involves the following steps (Schnoering & Vazirgiannis, 2025; Bellei et al., 2024; Meiklejohn et al., 2013; Harrigan & Fretter, 2016):

1. extracting the addresses involved in the transactions $\mathcal{T}$;
2. clustering the addresses into users using a heuristic $\mathcal{H}$ (or a combination thereof) applied to $\mathcal{T}$, potentially augmented with external information;
3. creating directed edges with associated features between users, derived from the $\mathcal{T}$;

4. generating node features by aggregating information from edges;
5. incorporating external information (off-chain) into both node and edge features.

**Hierarchical Clustering.** Hierarchical clustering constructs a hierarchy of nested clusters over a set of points $V$ endowed with a dissimilarity function $d$ (Heller & Ghahramani, 2005). In the agglomerative variant, each node initially forms its own cluster. At each step, two clusters $A, B \subset V$ are merged according to a linkage rule based on $d$. After the final step, all nodes are merged into a single cluster. This hierarchy is naturally represented by a rooted binary tree, or *dendrogram*, where leaves correspond to individual nodes, internal nodes represent successive merges, and node height indicates the merge distance. An example of dendrogram is illustrated in Figure 1.

**Graph Neural Networks (GNNs).** GNNs extend neural architectures to graph-structured data by propagating and transforming node features along edges. At each layer, a node updates its representation by aggregating information from its neighbors, allowing the model to capture both local connectivity and node attributes. By stacking multiple layers, GNNs learn embeddings that encode multi-hop structural context and can be used for tasks such as node classification, link prediction, and graph-level inference (Kipf, 2016; Hamilton et al., 2017; Veličković et al., 2017).

**Contributions.** The main contributions of this paper are threefold:

1. We publicly release a dataset of large-scale Bitcoin transaction graphs with a substantial number of clusters, enabling the training and evaluation of clustering algorithms at scale.
2. We propose a methodology for learning address embeddings consistent with traditional blockchain heuristics, supported by ~~theoretical guarantees and empirical validation~~ theoretical guiding intuitions and empirical analyses.
3. We show how these learned representations can refine heuristic-based clustering by ~~detecting and correcting cluster collapses~~ flagging potential cluster collapses, proposing candidate splits, and providing a hierarchical clustering that improves intelligibility and visualization.

## 2 Related Works

**Heuristics-Based Clustering.** To achieve address clustering, a variety of human-made, rule-based heuristics have been proposed (Schnoering et al., 2024), often based on behavioral patterns and human biases. The most prominent is the *common-input heuristic*, which assumes that all addresses providing inputs to the same transaction are controlled by a single entity. Clustering heuristics play a crucial role in Bitcoin analysis by approximating user-level structures from pseudonymous transaction data. They allow researchers and investigators to reduce complexity, uncover patterns of address ownership, and make sense of large-scale transaction graphs. Beyond their methodological value, such heuristics have become essential tools in several domains: in forensic contexts (Meiklejohn et al., 2013; Foley et al., 2019); in compliance and anti–money-laundering efforts (Möser et al., 2013; Yang et al., 2023), and in privacy research (Androulaki et al., 2013).

**Other Methods for Address Clustering.** Aside from heuristic clustering, other methods have been used on bitcoin transaction networks to similar tasks. Machine-learning based methods tend to focus more on the orthogonal task of address classification (Toyoda et al., 2018; Lin et al., 2019; Garin & Gisin, 2023; Sie et al., 2025; Jia et al., 2018; Lee et al., 2020), which consists in identifying the usage of addresses (e.g. scams, marketplaces). Some of those approaches (Kang et al., 2020) use heuristic clustering as a first step before training a classifier. More recently, approaches leverage GNNs to obtain powerful representation of transaction graphs for downstream tasks (Zhao et al., 2025; Zhang et al., 2025; Huang et al., 2022b).

**Enhancing Clustering Heuristics with GNNs.** Despite their usefulness, heuristic methods have notable limitations. They yield only *flat* cluster assignments—single-level groupings in which addresses are either linked or not—making large clusters difficult to interpret. Some heuristics also merge addresses based on a single transaction, which can erroneously combine unrelated users and cause cluster collapse (Androulaki

et al., 2013; Harrigan & Fretter, 2016). Only a few studies attempt to refine or correct the traditional heuristics. Möser & Narayanan (2022) use a random forest to estimate the likelihood that a heuristic-based merge is valid and block merges with low confidence, thereby mitigating cluster collapse. Similarly, Ermilov et al. (2017) uses off-chain information as votes for separating clusters.

Our method differs in key ways. Instead of assigning confidence scores to individual merges, we learn address embeddings that capture the global transaction structure while staying consistent with heuristic clusters. Agglomerative hierarchical clustering on these embeddings yields a dendrogram that reveals nested substructures and provides a principled criterion for detecting suspicious merges, producing both a refined flat clustering and a multi-resolution view of the address graph.

## 3 Methodology

### 3.1 Methodology Overview

We present a method to learn address embeddings consistent with standard heuristics, mapping nodes from the same cluster close together and pushing nodes from different clusters apart. These embeddings are then used to build dendrograms whose hierarchical structure reveals discrepancies in the heuristic partitions—most notably cases of cluster collapse—and to propose ~~corresponding corrections~~ candidate refinements. Throughout the paper, let $G = (V, E)$ denote the graph, where $V$ is the set of nodes (Bitcoin addresses) and $E$ the set of edges (value transfers). We write $\mathcal{C} = \{C_1, \ldots, C_k\}$ for a partition of $V$ (e.g., obtained via heuristics), with $k$ the number of clusters.

**Rationale for the Two-Stage Methodology.** Our approach is in line with a broad body of prior work and offers a key practical advantage: it naturally accommodates dynamic graphs with continuously arriving addresses and transactions, closely reflecting real-world blockchain conditions. In contrast, most end-to-end GNN pooling methods (Ying et al., 2018; Bianchi et al., 2020) construct a fixed hierarchy of merged nodes whose depth and cluster sizes are predetermined by the network architecture. Such constraints hinder adaptation to a continually growing transaction graph and reduce the interpretability of the resulting merges. Other pooling approaches (Lee et al., 2019) merely score and retain important nodes without producing a true hierarchical clustering, offering saliency rather than an interpretable dendrogram of successive merges.

### 3.2 Data Acquisition and Graph Construction

We construct our graphs using the pipeline of Schnoering & Vazirgiannis (2025)[1]. The procedure follows the steps outlined in the introduction—parsing the blockchain, extracting transactions, and forming entity-to-entity links—but, unlike the original work, we do not pre-cluster addresses into user entities. The resulting network is a directed graph with nodes as addresses. User clusters serving as ~~ground truth~~ heuristic training labels for supervised learning are obtained with the same set of address-clustering heuristics as in Schnoering et al. (2024), also implemented in the above GitHub repository. Constructing a graph from the entire history would yield billions of nodes and edges, rendering most algorithms intractable. We therefore sample a subset of transactions from a contiguous block interval to build the graph; the sampling strategy is described in the Appendix A.1.1. For complete implementation details, we refer readers to the original paper and accompanying code. The raw blockchain data for graph construction and clustering were obtained by running Bitcoin Core[2].

### 3.3 Learning Node Embeddings with GNNs and Contrastive Loss

We train a GNN $g$ to produce node embeddings consistent with the clustering $\mathcal{C}$: nodes within the same cluster (user) should have similar embeddings, whereas embeddings of nodes from different clusters should

---

[1] https://github.com/hugoschnoering2/BTCGraphConstruction
[2] https://bitcoin.org/en/bitcoin-core

be dissimilar. To enforce this, we adopt the contrastive InfoNCE loss (Oord et al., 2018; Chen et al., 2020)

$$\mathcal{L} = \mathbb{E}_{\mathbb{P}_\alpha}\left[-\log \frac{\exp\big(g(X)\cdot g(X^+)/\tau\big)}{\exp\big(g(X)\cdot g(X^+)/\tau\big) + \sum_{i=1}^{p}\exp\big(g(X)\cdot g(X_i^-)/\tau\big)}\right], \tag{1}$$

where $\mathbb{P}_\alpha$ is the sampling distribution over anchor nodes, $\tau$ is a temperature hyperparameter, and $p$ is the number of negative samples. For each anchor $X \in V$, the positive sample $X^+$ is drawn from the same cluster, while the negatives $\{X_i^-\}_{i=1}^{p}$ come from different clusters. Clusters are drawn from a mixture of uniform and size-proportional sampling controlled by $\alpha$, and nodes are then sampled uniformly within each chosen cluster. Our use of a contrastive objective is theoretically aligned with previous works (HaoChen et al., 2021; Tan et al., 2024) which proves that contrastive losses recover eigenvector-like representations of an underlying data graph, yielding cluster-separable embeddings under mild connectivity assumptions. Full details of this sampling scheme are provided in Appendix. Although the formula omits explicit normalization, we normalize embeddings in practice so that the dot product computes cosine similarity.

### 3.4 Detecting and ~~Correcting~~ Mitigating Potential Cluster Collapse

We perform agglomerative hierarchical clustering on the embeddings using cosine distance, consistent with the contrastive loss. Starting from the coarse partition $\mathcal{C}$, we cluster each $C_i$ independently, building a dendrogram that records the merge distances within every initial community.

Given a threshold $\lambda > 0$, ~~we define a *collapse* as any merge whose cosine distance exceeds $\lambda$~~ we flag as suspicious any merge whose cosine distance exceeds $\lambda$. This provides a principled way to flag suspicious merges—likely combining addresses from different users—and highlights potential failures of the original flat clustering. ~~To correct such collapses~~ To mitigate such potential collapses, we split the affected clusters into their hierarchical subcomponents, yielding a refined partition that may better reflect the true user structure.

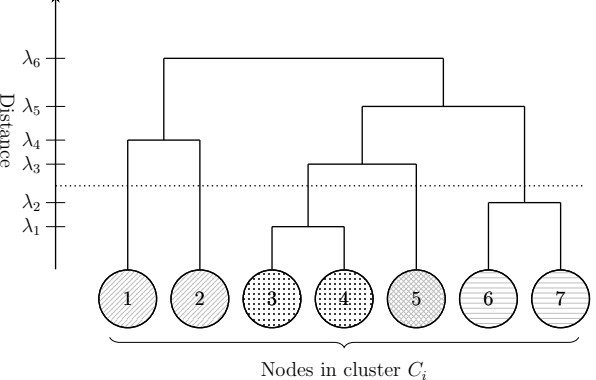

Figure 1: Example of a refinement. The dotted line represents the cut. Sub-clusters are distinguished by node fill patterns. ~~Merges above the threshold are treated as collapses~~ Merges above the threshold are flagged as potential collapses.

Mathematically, each dendrogram induces an ultrametric $d_u$ on the node set $V$, where $d_u(x,y)$ is the height of the lowest common ancestor of $x$ and $y$. Two nodes $x$ and $y$ are grouped together if they belong to the same initial cluster $C_i$ and satisfy $d_u(x,y) < \lambda$. This refinement process is illustrated in Figure 1.

A practical variant of this approach uses heuristic-generated clusters as the initial partition, motivated by the observation that such heuristics often merge distinct communities (i.e., distinct Bitcoin users).

## 4 ~~Theoretical Foundations~~ Theoretical Guiding Intuitions

~~We show that node embeddings learned by GNNs naturally separate nodes according to cluster membership in a hierarchical dendrogram, under appropriate conditions.~~ We provide a theoretical intuition for why GNN embeddings can separate nodes according to cluster membership in a hierarchical dendrogram under idealized conditions. Let $d$ be the working distance on $V$, and build a dendrogram from $d$ using single, average, or complete linkage. Assume the ground-truth clusters are well $d$-separated: there exist constants $0 < r < s$ such that $d(x,y) \le r < s \le d(x,z)$ for all $x,y \in C_\ell$ and every $z \in C_m$ with $\ell \neq m$. In other words, intra-cluster distances are uniformly smaller than inter-cluster distances. It then follows that any horizontal cut of this dendrogram at a threshold $\lambda \in (r,s)$ exactly recovers $C$; the resulting flat clustering coincides with the ground truth. Although these conditions are stronger than typically encountered in practice, ~~they~~

~~provide a clean theoretical framework for the analysis that follows~~ they provide a clean idealized framework for interpreting the analysis that follows and already motivate the use of a contrastive loss (HaoChen et al., 2021; Tan et al., 2024). The formal statements and derivations are provided in Appendix F.

**Notation.** Let $L$ be the (unnormalized) graph Laplacian of $G$, defined as $L = D - A$, where $A$ is the adjacency matrix of $G$ and $D$ is the diagonal degree matrix with entries $D_{ii} = \sum_j A_{ij}$. Let $\lambda_1 \leq \lambda_2 \leq \cdots \leq \lambda_n$ be its eigenvalues and $u_1, \ldots, u_n$ the associated orthonormal eigenvectors, which form an orthonormal basis of $\mathbb{R}^n$. Let $U \in \mathbb{R}^{n \times n}$ be the matrix whose columns are these eigenvectors. The spectral decomposition of $L$ is $L = UDU^\top$, where $D = \mathrm{diag}(\lambda_1, \ldots, \lambda_n)$ is the diagonal matrix of eigenvalues. Let $U_k \in \mathbb{R}^{n \times k}$ be the matrix formed by the first $k$ eigenvectors. For a node $i \in V$, its spectral embedding is $e_i^s = (u_{i,1}, u_{i,2}, \ldots, u_{i,k}) \in \mathbb{R}^k$, where $k$ is the number of clusters in the partition $C$. We write $\|x\|_2$ for the Euclidean norm of a vector $x$. For any matrix $A$, $A^\top$ denotes its transpose, $\sigma_{\min}(A)$ the smallest singular value of $A$, and $\|A\|_{\mathrm{op}}$ for the operator norm of $A$ induced by $\|\cdot\|_2$.

### 4.1 Results

Building on the perfect–cut criterion above, our goal is to derive a separability condition on the problem data that guarantees a dendrogram built from GNN embeddings admits such a perfect cut. Both results in this section assume that the working distance is Euclidean. The arguments, however, remain valid for cosine distance provided that the GNN embeddings lie on a common sphere. As a first step, Lemma 1 establishes an analogous condition for spectral embeddings. This intermediate result is natural because GNNs typically act as low-pass spectral filters (Nt & Maehara, 2019), so their embeddings concentrate in the subspace spanned by the Laplacian eigenvectors with the smallest eigenvalues, i.e., the classical spectral embeddings (Von Luxburg, 2007). The result involves the spectral distance between the Laplacian $L$ and the Laplacian $L^\circ$ of an *ideal cluster graph*, where two nodes are connected if and only if they belong to the same cluster. This ideal graph represents a perfectly homophilic scenario in which edges exist only within clusters. The appearance of this quantity is motivated by empirical observations on data, where addresses controlled by the same user tend to form connected subgraphs.

**Lemma 1.** *The spectral embeddings are cluster–separable whenever*

$$M := 4\sqrt{2k}\left(1 - \tfrac{1}{S_{\max}}\right)\|L - L^\circ\|_{\mathrm{op}} < \frac{1}{\sqrt{2S_{\max}}}.$$

*where $S_{\max}$ is the size of the largest cluster, and $L^\circ$ the Laplacian of the ideal cluster graph.*

The proof in Appendix F.1 relies on a version of the Davies–Kahan theorem from matrix perturbation theory. The separability condition is satisfied whenever the graph Laplacian $L$ is sufficiently close to the ideal block–diagonal Laplacian.

We assume that the node embeddings $H$ produced by the GNN can be written as $H = p(L)\,XW$, where $p$ is a polynomial, $X$ the matrix of initial node features, and $W$ the learned weight matrix (as in the linearized GCN (Kipf, 2016), for example). Using the spectral decomposition $L = UDU^\top$, this becomes

$$H = U\,\tilde{D}\,U^\top XW,$$

where $\tilde{D} = \mathrm{diag}(p(\lambda_1), \ldots, p(\lambda_n))$. The polynomial $p$ acts as a *spectral filter*, selectively amplifying or attenuating the eigencomponents of $L$ according to their eigenvalues. In the special case of an ideal low-pass filter, $p(\lambda_i) = \mathbf{1}_{\{i \leq k\}}$, so the embeddings lie entirely in the subspace spanned by the first $k$ eigenvectors. To measure how well a GNN approximates this ideal filter, we define $\alpha = \max_{i \leq k} |p(\lambda_i)|$, $\beta = \max_{i > k} |p(\lambda_i)|$, and $\gamma = \min_{i \leq k} |p(\lambda_i)|$. Theorem 2 transfers this spectral result to the ~~learned GNN embeddings, yielding an equivalent separability condition for the perfect cut—a result that, to our knowledge, is novel.~~ linearized GNN embeddings, yielding a sufficient separability condition for the existence of a perfect cut in this idealized setting.

**Theorem 2.** *The GNN embeddings are cluster–separable whenever*

$$\|XW\|_{\mathrm{op}}(\beta + \alpha M) \; < \; \gamma\,\sigma_{\min}(U_k^\top XW)\left(\sqrt{2/S_{\max}} - M\right).$$

The embeddings learned by the GNN inherit the geometric separability of the spectral embeddings, up to perturbations controlled by the low-pass approximation quality of $p$ and by the alignment of the feature matrix $XW$ with the leading eigenspace. Because the left-hand side of the inequality is positive, the separability condition can be satisfied only if three requirements are met: (i) $\gamma > 0$, so the GNN retains all eigencomponents of the informative subspace; (ii) $\sigma_{\min}(U_k^\top XW) > 0$, ensuring that the transformed features are not orthogonal to this subspace; and (iii) $M \leq \sqrt{2/S_{\max}}$, meaning the observed graph is sufficiently close to the ideal block-diagonal Laplacian so that spectral embeddings themselves already separate the clusters.

We emphasize that these results are not intended as directly verifiable guarantees on real Bitcoin transaction graphs. Rather, they formalize the geometric intuition that hierarchical refinement is meaningful when learned embeddings exhibit a separation between intra-cluster and inter-cluster distances.

### 4.2 Related Works

Spectral embeddings have long been central to graph clustering (Von Luxburg, 2007). Most theoretical analyses relate these embeddings to the *optimal* solutions of node-partitioning problems, including RatioCut minimization (Von Luxburg, 2007), $k$-way partitioning (Peng et al., 2015), and maximum-margin clustering (Hofmeyr, 2020). The guarantees in these works require the reference clustering to coincide with the optimal solution of the respective problem. Our approach makes no such assumption. We instead study graphs that are small perturbations of an *ideal cluster graph* whose connected components match the ground-truth clusters, and we apply matrix perturbation theory to obtain ~~our guarantees~~ sufficient separability conditions. This technique was also used by Ng et al. (2001) to bound intra-cluster variance. In contrast, we establish *pairwise* bounds—both within and across clusters—yielding separability conditions that ensure a perfect cut.

## 5 Experimental Setup

All experiments were performed on a Mac M3 Max equipped with 36 GB of RAM, using only CPU computation and no GPU acceleration.

We use the pipeline described in Section 3 to generate graphs from Bitcoin transactions. In total, we construct three graphs for training, one for validation, and one for testing. To avoid information leakage, transaction sets are sampled from non-overlapping block intervals so that no transaction appears in more than one graph. The main characteristics of these graphs are provided in Appendix A, and all datasets, including the graphs used in the experiments with ground truth labels in Section 6.3, are publicly available at *** under the CC BY 4.0 license.

Before being fed to the GNNs, features undergo the normalization and log-scaling procedure detailed in Appendix B.2. This step ensures consistent feature distributions across the different graphs.

### 5.1 Training

**Setup.** We train two-layer GNNs to minimize the contrastive loss of Equation 1, monitoring progress by evaluating the same loss on a validation graph. We experiment with three popular architectures: Graph Convolutional Network (GCN) (Kipf, 2016), GraphSAGE (Hamilton et al., 2017), and Graph Attention Network (GAT) (Veličković et al., 2017). Optimization uses Adam (Kingma & Ba, 2014) with a learning rate halved when the validation loss does not improve for 20 consecutive epochs. Because we have three training graphs, we cycle through them every 15 epochs to promote generalization. To accelerate training, we adopt neighborhood sampling (Hamilton et al., 2017), drawing 15 neighbors for the first GNN layer and 5 for the second. All experiments rely on the `PyTorch Geometric` implementations of GNN models, the Adam optimizer, learning-rate scheduler, and neighbor sampling. The code used in this study is publicly available at ***. Unless otherwise specified, all hyperparameters are listed in Table 6 of Appendix B.3.

**Model Variations.** We evaluate three main variations of the base model. (1) Because the constructed graphs are directed, we optionally symmetrize them before input to the GNN. (2) Since edges carry attributes, we can include or ignore these edge features whenever the architecture supports them. (3) We optionally

add a structural positional encoding to enhance locality. GNN message passing tends to make nodes with similar neighborhoods appear similar—even when they are far apart (Xu et al., 2019)—which can spuriously cluster structurally alike but unrelated nodes. Yet Bitcoin addresses belonging to the same user are usually close in the graph, as they often participate in the same transactions. To exploit this property, we follow the position-aware GNN framework (You et al., 2019): we select the highest-degree nodes as landmarks and represent each node by its vector of shortest-path distances to these landmarks. These distances are converted to similarities via $x \mapsto (1 + x)^{-1}$ and normalized dimension-wise. The resulting distance-based vector is then concatenated with the original node feature vector before message passing.

## 5.2 Evaluation

We evaluate our method by its ability to recover both hierarchical and flat clusterings consistent with the ~~ground truth~~ heuristic partition. For the hierarchical step, we apply agglomerative clustering with cosine distance on the GNN embeddings using average linkage. Because the graphs are large and computing the full pairwise distance matrix is impractical, we first obtain a coarse partition with the Leiden algorithm (Traag et al., 2019), limiting the maximum community size to 65 000 nodes to control memory usage, following the strategy described in Section 3.4.

**Metrics.** We score the resulting dendrograms with *dendrogram purity* (Heller & Ghahramani, 2005), which ranges from 0 to 1 and measures how well nodes from the same ~~ground-truth cluster~~ heuristic labels merge together. ~~Flat clusterings are obtained by cutting each dendrogram at a threshold $\lambda$ (Figure 1), selected by grid search to maximize the global silhouette score~~ (Rousseeuw, 1987~~), a standard criterion for choosing the cut level in hierarchical clustering.~~ Flat clusterings are obtained by cutting each dendrogram at a threshold $\lambda$ (Figure 1). This threshold is selected by first maximizing the silhouette score locally within each Leiden community (Rousseeuw, 1987), and then aggregating the resulting local thresholds through a size-weighted average. Full details are provided in Appendix C.2. We then compare the flat partition to the ~~ground truth~~ heuristic partition using Normalized Mutual Information (NMI) and Adjusted Rand Index (ARI) (Vinh et al., 2009): NMI captures global agreement and is robust to cluster-size imbalance, while ARI emphasizes local consistency but is more sensitive to class imbalance. Additional implementation details on how these metrics are computed, as well as their formal definitions, are provided in Appendix C.1. For all metrics we evaluate only nodes with degree $\geq 2$, excluding peripheral addresses that often lack sufficient transactional context for reliable user clustering and can artificially inflate cluster counts, making global metrics less informative.

**Baselines.** To highlight the added value of the contrastive loss, we compare our model to three unsupervised baselines: (i) an untrained GAT, (ii) a GAT trained as a non-probabilistic Graph Auto-Encoder (GAE) (Kipf & Welling, 2016), and (iii) a GAT trained with Deep Graph Infomax (DGI) (Veličković et al., 2018), which maximizes mutual information between local and global representations. All baselines produce node embeddings that are clustered exactly as in our contrastive pipeline. Implementation details are provided in Appendix B.5.

# 6 Results

## 6.1 Ablation Study

We report in Table 1 the performance results for different variations, including graph symmetrization, use of edge features, and the number of landmarks in the structural embedding. For each model variation, we averaged the results over five runs with different random seeds on the test graph. Additional experiments on the embedding dimension, the sampling parameter $\alpha$, and the number of negative anchors in the contrastive loss, as well as ~~empirical evidence that our method approaches the conditions required by the theoretical results~~ empirical indicators related to the theoretical intuitions, are reported in Appendix D.

All baselines achieve ARI scores well above zero—substantially better than random clustering—confirming that graph topology alone carries meaningful cluster information and supporting the homophily hypothesis. Louvain and Leiden remain significantly below all GNN-based approaches, both in NMI and ARI. The

| Model | Sym. | Edge feat. | # LM | DP | NMI | ARI |
|---|---|---|---|---|---|---|
| Louvain | ✓ | na | na | na | 0.642 (±0.000) | 0.289 (±0.000) |
| Leiden | ✓ | na | na | na | 0.665 (±0.000) | 0.311 (±0.000) |
| Random GAT | ✓ | × | 0 | 0.699 (±0.010) | 0.692 (±0.008) | 0.557 (±0.005) |
| GAE | ✓ | × | 0 | 0.738 (±0.008) | 0.755 (±0.011) | 0.609 (±0.015) |
| DGI | ✓ | × | 0 | 0.683 (±0.004) | 0.685 (±0.008) | 0.558 (±0.041) |
| GAT | × | × | 0 | 0.638 (±0.013) | 0.709 (±0.008) | 0.341 (±0.085) |
|  | × | × | 64 | 0.682 (±0.002) | 0.728 (±0.001) | 0.588 (±0.010) |
|  | × | × | 128 | 0.684 (±0.002) | 0.737 (±0.004) | 0.593 (±0.009) |
|  | × | × | 256 | 0.678 (±0.002) | 0.728 (±0.004) | 0.588 (±0.005) |
| GAT | × | ✓ | 0 | 0.635 (±0.005) | 0.707 (±0.006) | 0.344 (±0.049) |
|  | × | ✓ | 64 | 0.680 (±0.002) | 0.730 (±0.002) | 0.586 (±0.007) |
|  | × | ✓ | 128 | 0.683 (±0.005) | 0.733 (±0.004) | 0.585 (±0.005) |
|  | × | ✓ | 256 | 0.678 (±0.002) | 0.732 (±0.005) | 0.582 (±0.006) |
| GAT | ✓ | × | 0 | $\underline{0.771}$ (±0.004) | $\underline{0.760}$ (±0.015) | $\underline{0.637}$ (±0.007)$^*$ |
|  | ✓ | × | 64 | 0.789 (±0.004)$^{**}$ | 0.768 (±0.002)$^{**}$ | 0.627 (±0.003) |
|  | ✓ | × | 128 | 0.784 (±0.002) | 0.764 (±0.011) | 0.620 (±0.017) |
|  | ✓ | × | 256 | 0.785 (±0.002)$^*$ | 0.759 (±0.012) | 0.614 (±0.014) |
| GAT | ✓ | ✓ | 0 | 0.773 (±0.006) | 0.762 (±0.013) | 0.649 (±0.012)$^{**}$ |
|  | ✓ | ✓ | 64 | 0.782 (±0.004) | 0.756 (±0.007) | 0.609 (±0.006) |
|  | ✓ | ✓ | 128 | 0.778 (±0.003) | 0.756 (±0.013) | 0.611 (±0.019) |
|  | ✓ | ✓ | 256 | 0.780 (±0.002) | 0.755 (±0.010) | 0.612 (±0.011) |
| GCN | ✓ | × | 0 | 0.722 (±0.001) | 0.731 (±0.009) | 0.598 (±0.007) |
| GraphSAGE | ✓ | × | 0 | 0.765 (±0.003) | 0.766 (±0.011)$^*$ | 0.630 (±0.008) |

Table 1: Performance across different variations: graph symmetrization (Sym.), edge features (Edge feat.), number of landmarks (# LM), with evaluation metrics NMI, ARI, and dendrogram purity (DP), non applicable (na). The best score for each metric is marked with $^{**}$, the second-best with $^*$ and the performance of the model with all default parameters is underlined. All metrics in this table are computed with respect to heuristic-derived labels.

untrained GAT already substantially outperforms Louvain and Leiden, indicating that node features alone contain strong clustering signals. This is coherent with previous results showing that even untrained message passing models perform well on structured data (Huang et al., 2022a). Among unsupervised baselines, GAE achieves higher dendrogram purity (DP) than the random GAT and also improves NMI and ARI, suggesting that link-reconstruction objectives capture useful structural regularities. DGI performs comparably to the random GAT in DP and NMI, while exhibiting slightly more variable ARI scores, indicating less stable local separation.

When training GAT models on non-symmetrized graphs, performance deteriorates markedly across all metrics compared to the untrained GAT. This highlights the importance of reciprocal connectivity for capturing address relationships. Introducing structural positional encodings (landmarks) substantially improves performance in the non-symmetric setting. In contrast, incorporating edge features in the non-symmetrized case does not provide consistent gains and yields nearly identical results to the feature-free setting.

All symmetrized GAT variants outperform the baselines across metrics. The best overall dendrogram purity and NMI are achieved with 64 landmarks and no edge features. Interestingly, the highest ARI is obtained with symmetrization and edge features but without positional encoding, indicating that flat clustering quality may benefit from edge attributes even when hierarchical purity does not. Structural positional encodings consistently improve dendrogram purity in the symmetrized setting, though their impact on NMI and ARI is less systematic. The optimal number of landmarks appears to be 64, with diminishing or slightly negative returns beyond that. This suggests that moderate structural bias improves hierarchical coherence, whereas excessive positional information may introduce redundancy or overfitting effects.

Among alternative architectures, GraphSAGE slightly outperforms the default GAT in NMI and achieves competitive ARI, whereas GCN remains below. Nevertheless, the best overall performance across hierarchical and flat metrics is still obtained with the GAT architecture, supporting its use as the primary model in our framework.

## 6.2 Illustrating Cluster Refinement

~~We address potential cluster collapse using the procedure of Section 3~~ We illustrate how the procedure of Section 3 can flag and mitigate potential cluster collapse. Starting from the heuristic clustering, we

build a hierarchical clustering within each heuristic cluster and obtain a refined flat partition by cutting each dendrogram at the threshold $\lambda$ that maximizes the silhouette score. Figure 2 shows the resulting dendrogram for a representative cluster, with the selected cut level indicated. Its structure reveals the sequence of merges and highlights several late merges occurring above the optimal threshold. In particular, the final two subclusters merge at a cosine distance of 0.45, well above the chosen cut, indicating ~~that they should remain separate~~ a candidate split that may deserve further inspection. A few other merges also exceed the threshold, although most nodes merge below it into a single coherent group.

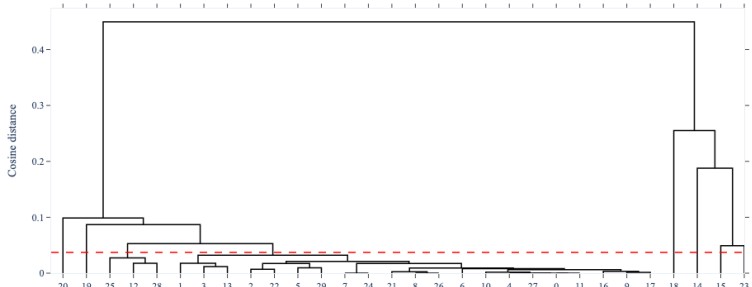

Figure 2: Dendrogram for a representative heuristic cluster. The dashed horizontal line indicates the cut level $\lambda$ selected to maximize the global silhouette score.

Figure 3 displays the minimal subgraph induced by the cluster and its neighbors. Cutting the dendrogram at the optimal threshold reveals coherent sub-groups, offering a clearer view of the cluster's internal organization. This approach naturally scales to much larger clusters, tens of thousands of nodes in our data and potentially millions in larger transaction sets, where direct graph visualization becomes impractical. Dendrograms provide a hierarchical, navigable representation that exposes meaningful substructures at multiple resolutions.

### 6.3 Additional Experiments with Independent Ground-Truth Labels

In each experiment, we focus on transactions for which ground-truth labels specify whether pairs of addresses belong to the same entity. For every labeled transaction, we extract a local transaction subgraph using the sampling procedure described in Section 3.2 (see Appendix A.1.2 for details), and construct the corresponding address-level graph. On each resulting graph, we compare three clustering strategies: (i) standard heuristic clustering, (ii) our default GNN–HAC pipeline, and (iii) a hybrid approach in which GNN embeddings refine the heuristic partition as described in Section 3.4. We further evaluate three linkage criteria and three dendrogram-cutting strategies (Appendix C.2).

In addition, we compare our approach to the refinement strategy of Möser & Narayanan (2022), which, to our knowledge, constitutes the only prior attempt at systematically refining heuristic-based address clustering. All implementation details required to reproduce this baseline, including our re-implementation and evaluation protocol, are provided in Appendix E.

Clustering quality is assessed using binary pairwise metrics: a prediction is correct if it groups addresses from the same entity or separates addresses from different entities, and incorrect otherwise. To prevent transactions with many labeled addresses from dominating the evaluation, we consider at most five randomly sampled labeled pairs per graph.

### 6.3.1 Entity Labels

We use the dataset of approximately 100,000 addresses labeled with entity names introduced in Schnoering & Vazirgiannis (2025). After excluding addresses associated with individuals, we sample 500 transactions between blocks 550,000 and 700,000 that involve at least two distinct labeled addresses. For each sampled transaction, addresses sharing the same entity label are expected to be assigned to the same cluster, whereas addresses associated with different entities should be placed in separate clusters. All experiments are re-

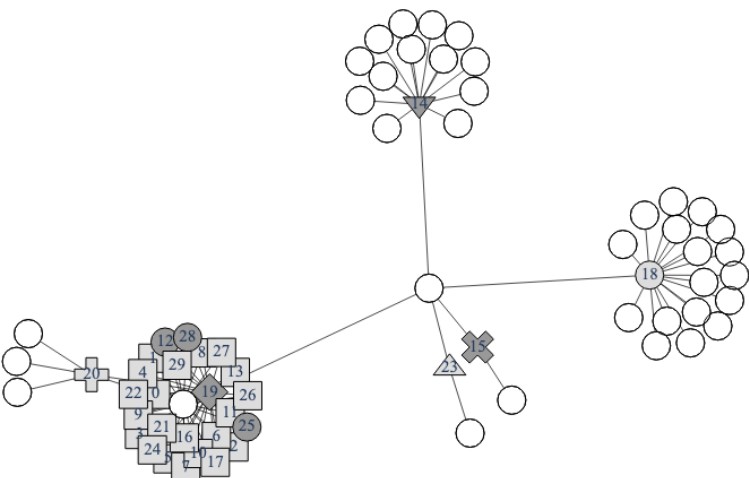

Figure 3: Minimal subgraph induced by the representative cluster and its immediate neighbors. Nodes belonging to the cluster are numbered, while external neighbors remain unnumbered. Cutting the dendrogram at the optimal threshold reveals distinct sub-groups, shown here with different gray shades and marker shapes.

| Model | Link. | Cut | TP(%) | FP(%) | FN(%) | TN(%) | bACC(%) | F1(%) |
|---|---|---|---|---|---|---|---|---|
| Heuristics | na | na | 42.6 (±0.3) | 22.7 (±0.2) | 15.8 (±0.5) | 18.9 (±0.6) | 59.2 (±0.7) | 59.2 (±0.7) |
| Möser & Narayanan (2022) | na | na | 41.2 (±0.1) | 22.4 (±0.2) | 17.8 (±0.3) | 18.6 (±0.3) | 57.6 (±0.2) | 57.6 (±0.2) |
| GNN-HAC | avg. | sil. | 38.0 (±0.4) | 16.7 (±0.9) | 20.3 (±0.6) | 25.0 (±0.7) | 62.6 (±0.6) | 62.3 (±0.5)[*] |
| GNN-HAC | avg. | inc. | 58.3 (±0.7) | 41.7 (±0.7) | 0.0 (±0.0) | 0.0 (±0.0) | 50.0 (±0.0) | 36.8 (±0.3) |
| GNN-HAC | avg. | gap. | 51.3 (±0.7) | 32.6 (±0.8) | 7.0 (±0.5) | 9.0 (±0.5) | 54.8 (±0.9) | 51.7 (±1.2) |
| GNN-HAC | ward | sil. | 34.5 (±1.4) | 15.5 (±1.0) | 23.8 (±1.8) | 26.2 (±0.8) | 61.0 (±0.9) | 60.4 (±1.1) |
| GNN-HAC | ward | inc. | 38.7 (±1.3) | 17.5 (±0.9) | 19.6 (±1.6) | 24.1 (±0.6) | 62.2 (±1.5) | 62.0 (±1.6) |
| GNN-HAC | ward | gap. | 49.3 (±0.8) | 27.3 (±1.3) | 9.0 (±0.9) | 14.4 (±0.9) | 59.5 (±1.0) | 58.6 (±1.3) |
| GNN-HAC | com. | sil. | 28.5 (±1.0) | 12.2 (±0.7) | 29.8 (±1.0) | 29.4 (±0.4) | 59.8 (±0.9) | 57.9 (±0.9) |
| GNN-HAC | com. | inc. | 58.3 (±0.7) | 41.6 (±0.7) | 0.1 (±0.1) | 0.1 (±0.1) | 50.0 (±0.0) | 37.0 (±0.3) |
| GNN-HAC | com. | gap. | 49.0 (±0.4) | 30.5 (±1.4) | 9.3 (±1.0) | 11.2 (±1.1) | 55.4 (±1.5) | 53.5 (±1.8) |
| Hybrid | avg. | sil. | 32.9 (±1.2) | 8.8 (±0.8) | 25.5 (±1.3) | 32.8 (±0.8) | 67.6 (±0.6)[**] | 65.7 (±0.8)[**] |
| Hybrid | avg. | inc. | 42.6 (±0.3) | 22.7 (±0.2) | 15.8 (±0.5) | 18.9 (±0.6) | 59.2 (±0.7) | 59.2 (±0.7) |
| Hybrid | avg. | gap. | 42.3 (±0.3) | 22.1 (±0.4) | 16.0 (±0.6) | 19.6 (±0.6) | 59.8 (±0.7) | 59.9 (±0.7) |
| Hybrid | ward | sil. | 34.7 (±0.6) | 16.1 (±0.7) | 23.6 (±1.2) | 25.6 (±0.8) | 60.5 (±1.2) | 60.0 (±1.3) |
| Hybrid | com. | sil. | 23.9 (±0.4) | 6.4 (±0.7) | 34.4 (±0.9) | 35.3 (±0.9) | 62.9 (±1.1)[*] | 58.7 (±1.2) |

Table 2: Clustering performance with ground-truth entity labels. Linkage (Link.) criteria: average linkage (avg.), Ward linkage (ward), and complete linkage (com.). Dendrogram cut methods: silhouette-based cut (sil.), inconsistency cut (inc.), and largest-gap cut (gap.). Metrics reported include true positives (TP), false positives (FP), false negatives (FN), true negatives (TN), along with balanced accuracy (bACC), defined as the mean of positive and negative recalls, and the macro-averaged F1 score (F1). Results are averaged over five random seeds corresponding to different random samples of labeled address pairs, with standard deviations reported in parentheses. The best score for bACC and F1 is marked with **, and the second-best with *.

peated over five random seeds, corresponding to different random samples of labeled address pairs used for evaluation. The results are reported in Table 2.

The average-linkage / silhouette-score configuration yields substantial improvements over the heuristic baselines. The best results are obtained with the refinement pipeline (Hybrid avg./sil.), underscoring the importance of the hybrid approach: macro-F1 increases from 59.2% to 65.7%, and balanced accuracy from 59.2% to 67.6%. In addition, the false-positive rate is reduced by more than half, from 22.7% to 8.8%, ~~effectively mitigating cluster collapse~~ suggesting that the refinement mitigates some false merges in this labeled setting. Complete linkage also strongly reduces false positives in the refinement setting, but this comes at the cost of a substantially higher false-negative rate and therefore lower macro-F1. In contrast, Ward linkage exhibits more mixed behavior: it performs competitively without refinement but does not benefit from the

| Model | Link. | Cut | TN(%) | TN(%) + CoinJoin negatives |
|---|---|---|---|---|
| Heuristics | na | na | 0.0 | 0.0 |
| GNN-HAC | avg. | sil. | 25.1 ($\pm$1.4) | 58.0 ($\pm$2.3) |
| GNN-HAC | ward | sil. | 43.9 ($\pm$1.5) | 57.9 ($\pm$1.9) |
| GNN-HAC | com. | sil. | 45.8 ($\pm$1.8) | 63.4 ($\pm$2.1) |
| Hybrid | avg. | sil. | 14.0 ($\pm$0.7) | 51.5 ($\pm$2.0) |
| Hybrid | ward | sil. | 11.4 ($\pm$0.3) | 30.2 ($\pm$0.5) |
| Hybrid | com. | sil. | 20.5 ($\pm$0.8) | 53.9 ($\pm$2.1) |

Table 3: Clustering performance with ground-truth CoinJoin labels. Results are reported as mean true-negative rate (TN) and standard deviation across five independently sampled CoinJoin graph datasets. Results are shown for average linkage (avg.), Ward linkage (ward), and complete linkage (com.) combined with silhouette-based dendrogram cuts (sil.).

hybrid correction step. The largest-gap criterion yields only modest improvements within the refinement pipeline. Finally, the inconsistency-based cut is generally not well suited to this task, as it either produces degenerate merge-heavy solutions with poor negative-class performance or collapses to behavior close to the heuristic baseline. Importantly, the best-performing configurations also outperform the refinement strategy of Möser & Narayanan (2022), demonstrating the added value of representation learning over rule-based merge filtering.

### 6.3.2 CoinJoin Transaction Labels

CoinJoin transactions involve multiple independent users and are explicitly designed to defeat the common-input heuristic (Schnoering & Vazirgiannis, 2025). Based on an analysis of open-source CoinJoin protocol implementations, Schnoering & Vazirgiannis (2023) proposed detection heuristics capable of identifying most such transactions. In a CoinJoin transaction, all input addresses are expected to belong to distinct entities and therefore should be assigned to different clusters. For each protocol examined in Schnoering & Vazirgiannis (2023), we randomly selected 100 transactions between blocks 550,000 and 700,000, totaling 500 CoinJoin transactions. Because these transactions contain only negative pairs (i.e., no two labeled input addresses should be clustered together), performance is evaluated solely through the true-negative rate. Classical heuristics without CoinJoin-aware safeguards, as reported in Schnoering et al. (2024), yield a true-negative rate of zero under this setting. To quantify sensitivity to the sampled CoinJoin set, we repeated only the evaluation-set sampling procedure five times with different random seeds, while keeping the trained model fixed, and report the mean and standard deviation across the five evaluations. Results for the different clustering methods are reported in Table 3.

~~Using the learned embeddings reduces the number of false positives induced by CoinJoin transactions. This effect is particularly pronounced in the setting without heuristic-cluster refinement, where complete linkage decreases the false-positive rate by approximately 25% relative to the baseline. In the refinement setting, however, the improvement remains limited. This may be due either to suboptimal threshold selection when cutting the dendrogram, or to the fact that the hierarchical structure does not sufficiently separate suspicious merges in these cases. Robustness to CoinJoin transactions could be further enhanced by explicitly incorporating such transactions as hard negative examples in the contrastive loss, thereby directly penalizing embeddings that place their input addresses close in representation space.~~

Embeddings learned with the standard contrastive objective of Equation 1 already improve robustness to CoinJoin-induced false positives. The best GNN-HAC configuration reaches a TN rate of 45.8% with complete linkage, compared to 0.0% for the heuristic baseline. The hybrid refinement setting also improves over the baseline, although its TN rates remain lower than those of the unconstrained GNN-HAC pipeline. To better understand these results, we analyze in Appendix D.4 the cosine distances between embeddings for CoinJoin negative pairs. This analysis shows that CoinJoin inputs often remain close in the embedding space under the standard contrastive objective. This is expected: input addresses of the same CoinJoin transaction are close in the transaction graph, yet they belong to distinct users. CoinJoin transactions therefore introduce strongly heterophilic patterns, with many independent users interacting within the same local transaction context. Our observations align with the findings of (Zhu et al., 2020), who demonstrate

that GNNs relying on homophily can misinterpret structurally proximity as semantic similarity. CoinJoin transactions exemplify this issue in Bitcoin graphs, since they deliberately mix unrelated addresses and thus violate homophily assumptions. This is further amplified by the divergence between the typical address connectivity and that of CoinJoin transactions, leading to negative transfer on those de-facto anomalous structure (Wang et al., 2024).

To adapt the model to this heterophilic failure mode, we augment the training objective with a CoinJoin-specific repulsion term, using CoinJoin input-address pairs as hard negatives. The resulting loss is $\mathcal{L} = \mathcal{L}_{\text{InfoNCE}} + \lambda_{\text{CoinJoin}}\mathcal{L}_{\text{CoinJoin}}$, where $\lambda_{\text{CoinJoin}}$ controls the weight of the CoinJoin repulsion term. This additional term penalizes high similarity between CoinJoin input embeddings and can be interpreted as a purely repulsive InfoNCE-like loss. Its full definition is provided in Appendix B.4, together with the hyperparameters used for the augmented training procedure. To avoid leakage from the evaluation set, we sampled a separate set of 1,000 CoinJoin-centered graphs used only for training. The corresponding results are also reported in Table 3. The augmented loss leads to substantially stronger CoinJoin robustness, with large TN gains across all linkage choices and particularly in the hybrid refinement setting. This demonstrates that the framework can naturally adapt once the relevant heterophilic patterns are incorporated into the supervision signal.

## Conclusion and Limitations

This work presents a principled framework for refining heuristic-based Bitcoin address clustering through contrastive GNN embeddings that remain consistent with standard heuristics while uncovering richer hierarchical structure. Starting from classical clustering rules, ~~our method learns embeddings that separate users in latent space and applies agglomerative hierarchical clustering to reveal substructures and flag suspicious merges~~ our method learns embeddings that encode heuristic-consistent similarity and applies agglomerative hierarchical clustering to reveal substructures and flag candidate suspicious merges.. Together, these elements provide a unified toolkit—data, theory, and methodology—for moving from flat heuristic clusters to interpretable, multi-resolution user graphs. A key limitation, however, is the limited amount of ground-truth labels available for evaluating user clusters at scale.

An important direction for future work is to adapt this procedure to a dynamic transaction graph that grows as new blocks and addresses appear, enabling online refinement of user clusters. A key challenge will be scalability. While node embeddings can be approximated by sampling subgraphs of manageable size, constructing the hierarchical structure is far less scalable, as illustrated in Appendix G. Nevertheless, this limitation can be partially mitigated by sampling very local subgraphs, either around specific transactions or within narrow temporal windows. This approximation is validated by the strong generalization to other thus sampled subgraphs observed in the empirical results. Future research should therefore focus on scalable hierarchical clustering techniques capable of handling continuously evolving blockchain graphs.

**Broader Impact.** Bitcoin address clustering has important dual-use implications. While it can support legitimate forensic, compliance, and anti-money-laundering applications, it may also weaken the practical pseudonymity of users who rely on address separation for privacy. All datasets and experiments in this work are derived from publicly available blockchain data and do not incorporate private off-chain information. Still, our results show that learned refinement methods can improve user-level clustering in some settings, which may affect privacy-seeking but legitimate users. We therefore frame this work as an analysis of the strengths and failure modes of heuristic clustering, including known privacy mechanisms such as CoinJoin, rather than as a production-ready deanonymization tool.

**LLM Usage** The research ideas, work and content presented in this paper was fully designed and made by the authors. LLMs were solely used in improving grammar and language a-posteriori, with no contribution besides reformulating for clarity purpose. In particular, no new idea or element was introduced through the use of an LLM.

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

# A Dataset

## A.1 Transactions Sampling Strategy

### A.1.1 Sampling from Coinbase Transactions

Constructing a graph from the full history of Bitcoin transactions would yield a network with several billions of nodes and edges, rendering most graph algorithms computationally infeasible. To obtain a manageable subgraph, we sample transactions occurring between two block indices $t_1 < t_2$.

A Bitcoin transaction transforms input value units into new output value units (TXOs), with unspent outputs known as UTXOs. Inputs originate from previous transactions, while outputs can be spent by future transactions. The only exception is the *coinbase transaction*—the first transaction in each block—which has no inputs and generates new currency units as a mining reward.

This structure naturally defines a directed acyclic graph (DAG): sources correspond to coinbase transactions; an edge exists from transaction $A$ to transaction $B$ whenever outputs from $A$ are consumed by $B$; sinks correspond to transactions whose no output has been spent. An example of such a transaction DAG is shown in Figure 4.

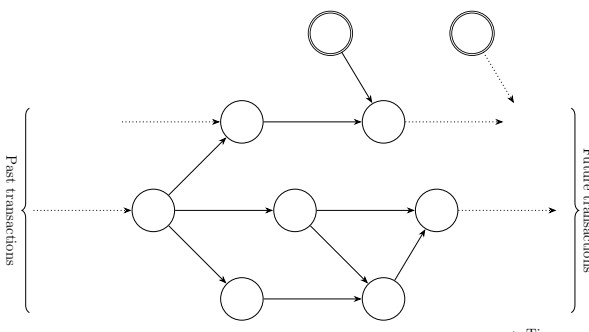

Figure 4: Bitcoin transaction graph. Circles represent transaction nodes, directed edges indicate the flow of bitcoin between transactions. Double–circled nodes denote coinbase transactions.

Our sampling procedure performs a breadth-first search (BFS) on this transaction DAG, initialized from a coinbase transaction chosen uniformly at random between blocks $t_1$ and $t_2$. Because block indices increase monotonically along transaction paths, all sampled transactions necessarily have indices greater than $t_1$, and exploration is truncated at block $t_2$, ensuring that every sampled transaction lies within the interval $[t_1, t_2]$. To further control the graph size, we cap the exploration depth at 15 and limit the number of transactions expanded at each BFS step to 5,000.

### A.1.2 Sampling from Transactions with Labels

In contrast to the directed exploration used for coinbase-based sampling, we perform a breadth-first search on the *undirected* transaction graph. This allows the procedure to explore not only future transactions consuming outputs of the seed, but also past transactions whose outputs were used as inputs to it.

To preserve locality, we use a maximum BFS depth of 3 and limit the number of expanded transactions per depth to 100. These constraints ensure that the sampled subgraph remains compact while capturing the relevant transactional context surrounding the labeled addresses. Aside from these modifications, the overall exploration logic follows the same structure as the coinbase-based sampling procedure described above in Appendix A.1.1.

## A.2 Graph Characteristics

Table 4 summarizes the key statistics of the sampled Bitcoin transaction graphs used for training, validation, and testing.

| Split | Blk int. | Blk dates int. | #Tx | #Nodes | #Edges | #Clust. |
|---|---|---|---|---|---|---|
| Train (1) | [599k, 600k] | 12/10/19 - 19/10/19 | 57k | 643k | 5.3M | 105k |
| Train (2) | [624k, 625k] | 02/04/20 - 08/04/20 | 48k | 491k | 3.7M | 96k |
| Train (3) | [674k, 675k] | 10/03/21 - 17/03/21 | 44k | 923k | 15.1M | 164k |
| Valid | [649k, 650k] | 19/09/20 - 26/09/20 | 57k | 828k | 7.0M | 137k |
| Test | [699k, 700k] | 04/09/21 - 11/09/21 | 56k | 1071k | 4.2M | 141k |

Table 4: Dataset statistics. Blk int. denotes the block interval, #Tx the number of sampled transactions, #Nodes the number of nodes, #Edges the number of edges, and #Clust. the number of clusters.

## A.3 Node and Edge Features

Table 5 lists the columns and their descriptions for each table (nodes, edges, and clusters) in the released graph dataset.

| Table | Column name | Description |
|---|---|---|
| Nodes | node_id | Identifier of the node |
| | degree_in | The number of incoming edges to the node |
| | degree_out | The number of outgoing edges from the node |
| | total_transaction_in | Total count of transfers received by the node |
| | total_transaction_out | Total count of transfers initiated by the node |
| | first_transaction_in | Block index of the first transfer received |
| | last_transaction_in | Block index of the last transfer received |
| | first_transaction_out | Block index of the first transfer sent |
| | last_transaction_out | Block index of the last transfer sent |
| | min_sent | Smallest value sent out in a single transaction |
| | max_sent | Largest value sent out in a single transaction |
| | total_sent | Cumulative value of all outgoing transfers |
| | min_received | Smallest value received in a single transaction |
| | max_received | Largest value received in a single transaction |
| | total_received | Cumulative value of all incoming transfers |
| Edges | a | Node id of the sender |
| | b | Node id of the recipient |
| | reveal | Block index of the first transaction |
| | last_seen | Block index of the last transaction |
| | total | Total number of transactions |
| | min_sent | Minimum sent in a single transaction |
| | max_sent | Maximum sent in a single transaction |
| | total_sent | Total sent in a single transaction |
| Clusters | node_id | Identifier of the node |
| | alias | Identifier of the cluster |

Table 5: Description of the columns of the different tables constituting the graph.

# B Training

## B.1 Sampling Function for Contrastive Learning

We assume a ~~ground-truth~~ reference clustering $\mathcal{C} = \{C_1, \ldots, C_k\}$ over the node set $V$. Let the latent variables $(Z, Z_1^-, \ldots, Z_p^-)$ denote the cluster labels of $(X, X_1^-, \ldots, X_p^-)$ under $\mathcal{C}$. The joint sampling distribution of equation 1 is

$$\mathbb{P}_\alpha(x, x^+, x_1^-, \ldots, x_p^-) = \sum_{z, z_1^-, \ldots, z_p^-} \mathbb{P}_\alpha(z) \, \mathbb{P}(x \mid z) \, \mathbb{P}(x^+ \mid z, x) \prod_{i=1}^{p} \mathbb{P}_\alpha(z_i^- \mid z) \, \mathbb{P}(x_i^- \mid z_i^-).$$

where $\mathbb{P}_\alpha(Z = z) = \alpha \frac{|C_z|}{|V|} + (1 - \alpha) \frac{1}{|C|}$ is a mixture between size-proportional sampling ($\alpha = 1$), $\mathbb{P}(X = x \mid Z = z)$ is uniform over all nodes in cluster $C_z$, $\mathbb{P}(X^+ = x^+ \mid Z = z, X = x)$ is uniform over $C_z \setminus \{x\}$, $\mathbb{P}_\alpha(Z_i^- = z_i^- \mid Z = z)$ is uniform over all $z_i^- \neq z$ and $\mathbb{P}(X_i^- = x_i^- \mid Z_i^- = z_i^-)$ is uniform over all nodes in cluster $C_{z_i^-}$.

This scheme provides a principled sampling strategy for contrastive learning: positive pairs are always drawn from the same cluster as the anchor, while negatives come from different clusters. The parameter $\alpha$ balances diversity and representativeness by interpolating between uniform and size-proportional cluster sampling.

## B.2 Features Preprocessing

Some input features encode amounts denominated in bitcoins (Table 5). Because the bitcoin price varies substantially across graph samples, we augment these features with their corresponding U.S.-dollar values, computed from the bitcoin price at each graph's starting date.

Feature preprocessing is performed independently for each graph. First, all features are log-transformed using $x \mapsto \log(1 + x)$ to reduce skewness. Next, we apply min–max normalization based on the empirical 5th and 95th percentiles of each feature, and missing values are imputed with zeros.

## B.3 Hyperparameters

Table 6 summarizes the model architecture, preprocessing options, and optimization settings used for training the GNNs.

|  | Hyperparameter | Value |
|---|---|---|
| Model | Number of attention heads | 4 |
|  | Size of hidden embeddings | 64 |
|  | Size of output embeddings | 128 |
|  | Number of layers | 2 |
|  | Activation function | Leaky ReLU |
|  | Dropout | 0.2 |
| Preprocessing | Symmetrize the input graph | True |
|  | Use edge features | False |
|  | Number of landmarks in the positional encoding | 0 |
| Optimizer | Initial learning rate | $2.5 \times 10^{-3}$ |
|  | Weight decay | $10^{-5}$ |
| Learning rate scheduler | Reduction factor | 0.5 |
|  | Patience | 20 |
| Gradient descent | Number of epochs | 250 |
|  | Num anchors per batch | 512 |
|  | Num negative samples per anchor ($p$) | 4 |
|  | Temperature ($\tau$) | 0.07 |
|  | Parameter of sampling function ($\alpha$) | 0.5 |

Table 6: Hyperparameters used in the training.

## B.4 CoinJoin Hard-Negative Training

Let $\mathcal{G}$ denote the set of available CoinJoin-centered training graphs. At each training step, we sample $N_g$ graphs from $\mathcal{G}$. For each sampled graph, we sample $N_p$ pairs of distinct input nodes involved in the CoinJoin transaction used to generate that graph. We denote these pairs by $(A_{g,j}, I_{g,j})_{j=1}^{N_p}$. The CoinJoin repulsion loss is defined as

$$\mathcal{L}_{\text{CoinJoin}} = \mathbb{E}\left[ \frac{1}{N_g} \sum_{g=1}^{N_g} \frac{1}{N_p} \sum_{j=1}^{N_p} \log\left(1 + \exp\left(\frac{g(A_{g,j}) \cdot g(I_{g,j})}{\tau}\right)\right) \right], \tag{2}$$

where the expectation is over the sampling of the CoinJoin-centered graphs and negative pairs.

The full augmented training objective is then

$$\mathcal{L}_{\text{InfoNCE}} + \lambda_{\text{CoinJoin}}\mathcal{L}_{\text{CoinJoin}}, \tag{3}$$

where $\lambda_{\text{CoinJoin}}$ controls the weight of the CoinJoin repulsion term.

The hyperparameters specific to this auxiliary CoinJoin loss are reported in Table 7.

| Hyperparameter | Value |
|---|---|
| Temperature ($\tau_{\text{CoinJoin}}$) | 0.30 |
| Weight ($\lambda_{\text{CoinJoin}}$) | 1.0 |
| Number of CoinJoin-centered graphs sampled per step ($N_g$) | 50 |
| Maximum number of negative pairs sampled per graph ($N_p$) | 32 |

Table 7: Hyperparameters used for the CoinJoin hard-negative training.

## B.5 Training of Baselines

**Louvain.** We use the Louvain implementation from the `NetworkX` library. The resolution parameter, which controls the granularity of the detected communities, is tuned by grid search in the range $[0.5, 3.0]$ on the validation graph to maximize the modularity score.

**Leiden.** For Leiden we rely on the `leidenalg` package, using the `RBConfigurationVertexPartition` objective (the standard modularity-based configuration). The resolution parameter is likewise tuned by grid search in the range $[0.5, 3.0]$ on the validation graph, and the number of refinement iterations is fixed to 10 to ensure convergence.

**Untrained GAT.** We follow exactly the same procedure as for the trained GNN experiments—using the default hyperparameters of Table 6—except that the number of training epochs is set to zero.

**Graph AutoEncoder (GAE).** We follow the non-probabilistic graph auto-encoder training procedure of Kipf & Welling (2016). The encoder is a GAT with the default hyperparameters of Table 6, while the decoder is a simple dot product. Given adjacency matrix $A$ and encoder embeddings $H$, the loss is the binary cross-entropy

$$\mathcal{L}_{\text{GAE}} = -\sum_{i,j}\big[A_{ij}\log\sigma(h_i^\top h_j) + (1 - A_{ij})\log\big(1 - \sigma(h_i^\top h_j)\big)\big],$$

where $\sigma$ is the sigmoid function. Embeddings are trained to reconstruct $A$. We apply the same neighbor sampling, training-graph rotation, and learning-rate scheduling as in the main experiments, monitoring performance via the validation-graph reconstruction loss. The only change in hyperparameters is a shorter training duration of 20 epochs.

**Deep Graph Infomax (DGI).** We adopt the training procedure of Veličković et al. (2018) for Deep Graph Infomax. The encoder is a GAT with the default hyperparameters of Table 6. DGI learns node embeddings by maximizing mutual information between local node representations and a global summary vector. Given node embeddings $H$ and a readout summary $s = \sigma\big(\frac{1}{n}\sum_i h_i\big)$, a corrupted graph $\tilde{G}$ is produced by randomly shuffling node features to create negative samples $\tilde{H}$. The loss is the binary cross-entropy

$$\mathcal{L}_{\text{DGI}} = -\sum_i\big[\log\sigma(h_i^\top W s) + \log\big(1 - \sigma(\tilde{h}_i^\top W s)\big)\big],$$

where $W$ is a trainable scoring matrix and $\sigma$ the sigmoid function. We use the same neighbor sampling, rotation of training graphs, and learning-rate scheduling as in the main experiments, and monitor training with the DGI objective on the validation graph. Training is limited to 20 epochs to match the GAE baseline.

## C   Evaluation

### C.1   Evaluation Metrics

We evaluate hierarchical and flat clusterings using standard information–theoretic and pairwise similarity measures.

#### C.1.1   Hierarchical Clustering

**Dendrogram Purity.**   Following Heller & Ghahramani (2005), let $T$ be a dendrogram with leaves $1, \ldots, n$ and class labels $c_1, \ldots, c_n$. To compute the purity of $T$:

1. Sample a leaf $\ell$ uniformly at random.
2. Sample another leaf $j$ uniformly at random among those with the same class label, $c_j = c_\ell$.
3. Let $S(\ell, j)$ be the smallest subtree of $T$ containing both $\ell$ and $j$.
4. Compute the fraction of leaves in $S(\ell, j)$ that share the class $c_\ell$.

The expected value of this fraction over the sampling procedure defines the *dendrogram purity*, which equals 1 if and only if every ground-truth class forms a pure subtree.

*Implementation.* We provide an open-source implementation in our public repository. Purity is estimated by Monte-Carlo with $N = 10{,}000$ sampled pairs $(\ell, j)$. Each pair is drawn *within the same coarse Leiden cluster*; because this Leiden partition is identical across all evaluations, this sampling constraint does not introduce bias.

#### C.1.2   Flat Clustering

**Normalized Mutual Information (NMI).**   The uncertainty of a clustering is quantified by its *entropy*, $H(U) = -\sum_u p(u) \log p(u)$, where $p(u)$ is the probability of cluster $u$. The similarity between two clusterings $U$ and $V$ can then be measured by their *mutual information*, $I(U; V) = \sum_{u,v} p(u, v) \log \frac{p(u,v)}{p(u)p(v)}$, which captures how much knowing $V$ reduces the uncertainty of $U$. The NMI score normalizes mutual information to the range $[0, 1]$ via

$$\mathrm{NMI}(U, V) = \frac{2\, I(U; V)}{H(U) + H(V)}.$$

Because mutual information captures the overall dependency between the two label distributions, this normalization measures *global agreement* between entire clusterings rather than only local pairwise matches. Moreover, the ratio form compensates for differing cluster entropies, making the score *robust to cluster-size imbalance* and directly comparable across datasets of varying class distributions.

**Adjusted Rand Index (ARI).**   To assess pairwise agreement, the *Rand index* is defined as $\mathrm{RI} = \frac{a+b}{\binom{n}{2}}$, where $a$ denotes the number of element pairs assigned to the same cluster in both $U$ and $V$, and $b$ denotes the number of pairs assigned to different clusters in both. The Rand index is corrected for chance agreement with

$$\mathrm{ARI} = \frac{\mathrm{RI} - \mathbb{E}[\mathrm{RI}]}{\max(\mathrm{RI}) - \mathbb{E}[\mathrm{RI}]},$$

placing the score in $[-1, 1]$ and emphasizing local consistency.

*Implementation.* All NMI and ARI computations use the standard implementations from `sklearn`.

### C.2   Dendrogram Cutting Methods

Let $\mathcal{C} = \{C_1, \ldots, C_K\}$ be a coarse partition. For each cluster $C_k$ with $n_k = |C_k| \geq n_{\min}$, we compute cosine distances between embeddings $H_{C_k}$ and build a hierarchical clustering linkage matrix.

Each cutting rule produces a local threshold $t_k$. We define a global threshold by size-weighted averaging:

$$\lambda = \frac{\sum_{k:\, n_k \geq n_{\min}} n_k\, t_k}{\sum_{k:\, n_k \geq n_{\min}} n_k}.$$

**Silhouette-based cut.** For a given cut level producing a flat partition $\hat{y}$ of $C_k$, we evaluate its quality using the silhouette score. Let $D_k$ denote the cosine distance matrix within $C_k$. For each point $i$, define

$$a(i) = \frac{1}{|C(i)| - 1} \sum_{j \in C(i),\, j \neq i} D_k(i, j),$$

the average intra-cluster distance, and

$$b(i) = \min_{C' \neq C(i)} \frac{1}{|C'|} \sum_{j \in C'} D_k(i, j),$$

the minimal average distance to another cluster. The silhouette coefficient of $i$ is

$$s(i) = \frac{b(i) - a(i)}{\max\{a(i), b(i)\}},$$

and the global silhouette score is

$$\mathrm{Sil}(D_k, \hat{y}) = \frac{1}{n_k} \sum_{i \in C_k} s(i).$$

The selected cut is the one maximizing $\mathrm{Sil}(D_k, \hat{y})$.

**Largest-gap cut.** Let $h_{k,1}, \ldots, h_{k,n_k-1}$ denote the merge heights in the dendrogram of $C_k$, and let $h_{k,(1)} \leq \cdots \leq h_{k,(n_k-1)}$ be their sorted values. Define the consecutive gaps

$$\Delta_i = h_{k,(i+1)} - h_{k,(i)}, \qquad i = 1, \ldots, n_k - 2.$$

Let

$$i^\star = \arg\max_i \Delta_i.$$

The cut level is chosen as the midpoint of the largest gap,

$$h_{k,(i^\star)} + \frac{1}{2}\Delta_{i^\star}.$$

**Inconsistency-based cut.** The inconsistency coefficient quantifies how unusually large a merge height is compared to merges occurring below it in the dendrogram. It is computed by normalizing the difference between the merge height and the mean height of descendant merges by their standard deviation. High values indicate potentially spurious merges.

We select the threshold adaptively as $\mu_v + \alpha\sigma_v$, where $\mu_v$ and $\sigma_v$ are the empirical mean and standard deviation of the inconsistency coefficients within the cluster, and $\alpha > 0$ controls the sensitivity of the split.

## D  Additional Experiments

### D.1  Embedding dimension

A critical design choice is the number of dimensions in the embedding space produced by the GNN. If the dimensionality is too low, the model cannot adequately separate the numerous clusters. Conversely, a very high dimensionality increases algorithmic complexity and computational cost, and may even lead to dimensional collapse (Jing et al., 2021). To investigate this trade-off, we experimented with different

| Embedding dimension | DP | NMI | ARI |
|---|---|---|---|
| 16 | 0.717(±0.004) | 0.718(±0.005) | 0.588(±0.019) |
| 32 | 0.737(±0.004) | 0.731(±0.005) | 0.605(±0.003) |
| 64 | 0.770(±0.010) | 0.753(±0.010) | 0.631(±0.019) |
| 128 | 0.779(±0.005) | 0.766(±0.016) | 0.644(±0.018) |
| 256 | 0.789(±0.004)* | 0.773(±0.023)* | 0.647(±0.040)* |
| 512 | 0.798(±0.005)** | 0.777(±0.009)** | 0.665(±0.025)** |

Table 8: Performance across different embedding dimensions with evaluation metrics NMI, ARI, and dendrogram purity (DP). The best score for each metric is marked with ** and the second-best with *. All metrics are computed on the test graph / heuristic clustering, and results are averaged over five runs.

embedding sizes, and compared their performance in Table 8. For consistency, we set the number of hidden dimensions in the GAT to $\frac{2 \times \text{Size of output embedding space}}{\text{Number of attention heads}}$.

Model performance improves consistently as the embedding dimension increases across all three metrics. Dendrogram purity (DP) exhibits a strictly monotonic rise from 16 to 512 dimensions, indicating progressively better hierarchical separation. Both NMI and ARI follow the same upward trend, with the highest scores obtained at 512 dimensions. While intermediate dimensions (64–128) already yield strong performance, larger embeddings further enhance both global agreement (NMI) and local consistency (ARI). These results suggest that, in our setting, increasing the embedding dimensionality does not induce degradation and instead provides additional representational capacity that translates into measurable clustering gains.

## D.2 Parameter of the sampling function

In this section, we evaluate the influence of the sampling parameter $\alpha$ on the performance of our methodology. The sampling parameter $\alpha$ controls the balance between uniform and size-proportional cluster selection in the contrastive sampling distribution. Table 9 reports the results. The results reveal a clear dependence on the sampling parameter $\alpha$. ARI is maximized at $\alpha = 0$, indicating that uniform cluster sampling favors stronger local pairwise consistency. In contrast, both dendrogram purity and NMI peak at $\alpha = 0.2$, suggesting that a slight bias toward size-proportional sampling improves both hierarchical coherence and global agreement with the ground truth. For larger values of $\alpha$ ($\geq 0.6$), performance gradually degrades, particularly in terms of NMI and ARI, reflecting a loss of local discriminative power. Overall, small but non-zero values of $\alpha$ provide the best trade-off, balancing global structure preservation and local cluster separability.

| $\alpha$ | DP | NMI | ARI |
|---|---|---|---|
| 0. | 0.761(±0.003) | 0.759(±0.007) | 0.685(±0.017)** |
| 0.2 | 0.780(±0.004)** | 0.766(±0.003)** | 0.645(±0.002)* |
| 0.4 | 0.774(±0.007) | 0.759(±0.010) | 0.621(±0.008) |
| 0.6 | 0.776(±0.007) | 0.761(±0.006)* | 0.628(±0.014) |
| 0.8 | 0.777(±0.010)* | 0.750(±0.009) | 0.631(±0.008) |
| 1.0 | 0.774(±0.005) | 0.738(±0.014) | 0.614(±0.018) |

Table 9: Performance across different $\alpha$ with evaluation metrics NMI, ARI, and dendrogram purity (DP). The best score for each metric is marked with ** and the second-best with *. All metrics are computed on the test graph / heuristic clustering, and results are averaged over five runs.

## D.3 Number of negative samples

In this section, we evaluate the impact of the number of negative examples per anchor used in the contrastive loss on the performance of our methodology. The results are reported in Table 10. Increasing the number of negative samples $p$ generally improves hierarchical clustering quality. Dendrogram purity (DP) rises almost monotonically from $p = 1$ to $p = 64$, reaching its maximum at $p = 64$, indicating that stronger contrastive separation enhances hierarchical structure. NMI follows a similar upward trend, also peaking at $p = 64$, suggesting improved global agreement with the ground truth as more negatives are incorporated. In contrast, ARI is highest at $p = 1$ and remains relatively stable thereafter, with only minor variations across larger

values of $p$. Overall, larger numbers of negatives benefit hierarchical coherence and global structure recovery, while local pairwise consistency appears comparatively insensitive to $p$ beyond very small values.

| $p$ | DP | NMI | ARI |
|---|---|---|---|
| 1 | $0.764(\pm0.004)$ | $0.749(\pm0.014)$ | $0.641(\pm0.008)^{**}$ |
| 4 | $0.773(\pm0.005)$ | $0.755(\pm0.006)$ | $0.637(\pm0.010)$ |
| 16 | $0.779(\pm0.004)^{*}$ | $0.758(\pm0.005)$ | $0.637(\pm0.012)$ |
| 32 | $0.777(\pm0.005)$ | $0.761(\pm0.006)^{*}$ | $0.636(\pm0.008)$ |
| 64 | $0.781(\pm0.004)^{**}$ | $0.771(\pm0.010)^{**}$ | $0.639(\pm0.013)^{*}$ |

Table 10: Performance across different $p$ with evaluation metrics NMI, ARI, and dendrogram purity (DP). The best score for each metric is marked with $^{**}$ and the second-best with $^{*}$. All metrics are computed on the test graph / heuristic clustering, and results are averaged over five runs.

### D.4 Approaching the Theoretical Conditions

**Cluster homophily.** Ideally, homophily would be measured by the spectral norm $\|L - L^\circ\|_{\mathrm{op}}$ between the graph Laplacian $L$ and the ideal block-diagonal Laplacian $L^\circ$, but this is infeasible for graphs with millions of nodes. As a practical alternative we use the *cut ratio*, the fraction of edges that cross between clusters: a low cut ratio indicates that most edges remain inside clusters and thus reflects strong homophily. On the validation graph the overall cut ratio is 87%; restricted to subgraphs of size 10–100 it is 77%, for size 100–1000 it is 51%, and for size 1000–5000 it is 49%. To assess the significance of these scores given the graph topology, we randomly permuted 1% of node labels and recomputed the cut ratio over 300 trials. For each case we calculated a z-score as the difference between the original score and the mean of the permuted scores, divided by their standard deviation. The corresponding p-value is the empirical probability that a random permutation yields a clustering more homophilic than the original. The resulting z-scores are -9.42 (global), -3.04 (10-100), -1.09 (100-1000), and -1.49 (1000–5000), with all p-values below 0.01, confirming that the observed homophily is highly significant for the graph topology.

**Low-pass GNN behavior.** For embeddings $H^{(\ell)}$ at layer $\ell$, the Dirichlet energy $\mathcal{E}(H^{(\ell)}) = \mathrm{Trace}\big((H^{(\ell)})^\top L H^{(\ell)}\big) = \sum_{i=1}^{n} \lambda_i \|H_i^{(\ell)}\|_2^2$ measures the concentration of $H^{(\ell)}$ on high-frequency eigenvectors. Normalizing by total energy gives the Rayleigh quotient $R(H^{(\ell)}) = \mathrm{Trace}\big((H^{(\ell)})^\top L H^{(\ell)}\big)/\mathrm{Trace}\big((H^{(\ell)})^\top H^{(\ell)}\big)$. A GNN acting as a low-pass filter should yield small Rayleigh quotients that decrease across layers. Across five training runs of a two-layer GAT with default parameters, the Rayleigh quotient decreases from 6.54 for the input embeddings $H^{(0)}$ to 1.29 after the first convolution $H^{(1)}$, then to 1.20 after the first activation (still $H^{(1)}$), and finally to 0.99 at the output $H^{(2)}$ on the validation subgraph. This monotonic drop confirms the expected low-pass filtering behavior.

**Embedding-Distance Distributions and Separability.** The theoretical analysis in Section 4 relies directly on a separation condition between intra-cluster and inter-cluster distances: pairs belonging to the same cluster should be close in the embedding space, while pairs belonging to different clusters should be farther apart. To empirically assess this intuition, we compare several distributions of cosine distances between embeddings, reported in Figure 5: positive and negative pairs derived from the heuristic labels used during training, positive and negative pairs from the independent entity-label evaluation, and negative pairs induced by CoinJoin transactions.

This separability is very clear for pairs defined by the heuristics. Heuristic positive pairs are strongly concentrated at small distances, whereas heuristic negative pairs are shifted toward much larger distances. This confirms that the contrastive objective learns the geometry it is explicitly trained to produce: pulling together addresses from the same heuristic cluster and pushing apart addresses from different heuristic clusters. The separation is less pronounced for pairs derived from independent entity labels, which is expected since these labels are not used as direct supervision during training and may reflect a noisier structure that is not perfectly aligned with the heuristics. Nevertheless, the separation remains visible: entity-label positives tend to be closer than entity-label negatives. This suggests that the embeddings do not merely reproduce heuristic labels mechanically, but partially transfer to an independent notion of entity identity. This empirical separation also provides a geometric explanation for the strong results obtained in Table 2.

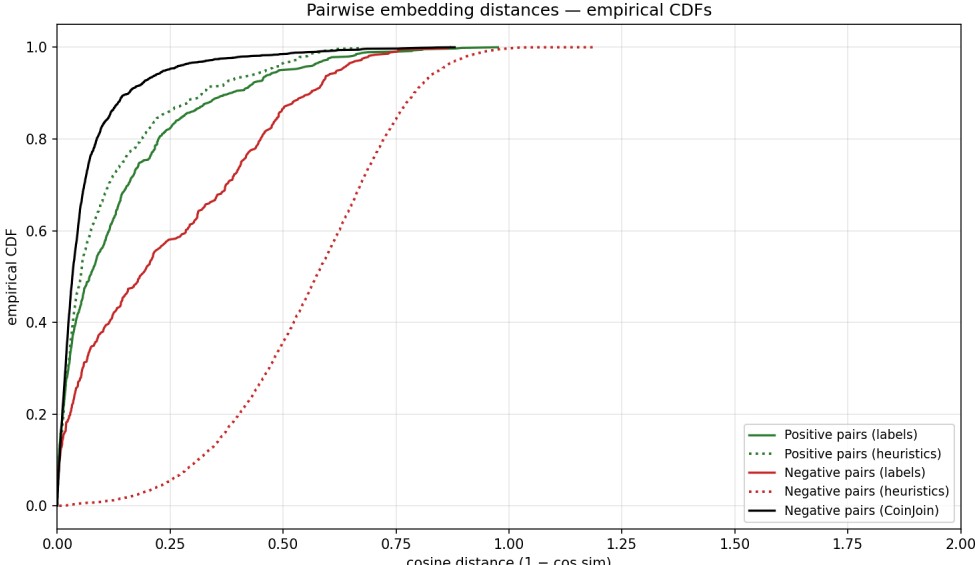

Figure 5: Empirical CDFs of cosine distances between embedding pairs without CoinJoin hard negatives.

CoinJoin negative pairs exhibit a different behavior, as shown in Figure 5. Without CoinJoin hard-negative training, they remain concentrated at much smaller distances than ordinary negative pairs, and their distribution is closer to the positive-pair distributions. This shows that the standard contrastive objective does not sufficiently separate CoinJoin inputs in the embedding space. This behavior is consistent with the structure of CoinJoin transactions: many independent users appear in the same local transaction context. The graph therefore induces strong structural proximity, although the corresponding addresses belong to distinct users. CoinJoin transactions thus create a strongly heterophilic pattern, where proximity in the graph does not correspond to common ownership.

After adding the CoinJoin repulsion loss, Figure 6 shows that the distance distribution of CoinJoin negative pairs shifts markedly toward larger values. The model therefore learns to better separate these hard negatives, and their distribution becomes closer to that of ordinary negative pairs. This provides a geometric explanation for the improved true-negative rates reported in Table 3, and shows that the framework can adapt once the relevant heterophilic patterns are explicitly incorporated into the supervision signal.

## E   Reproducing the Moser and Narayanan Heuristic

In this section, we reproduce the heuristic-refinement methodology introduced by Möser & Narayanan (2022). The methodology focuses on payment transactions with exactly two outputs: one corresponding to the payment itself, and the other returning change, defined as the excess input value relative to the payment amount. Under common privacy practices, it is often impossible to determine which output represents the payment and which represents the change. Consequently, a number of heuristics have been proposed to infer the change address based on recurring behavioral patterns. Möser & Narayanan (2022) employ twenty-six such heuristics. Given a dataset of payment transactions with two outputs, Möser & Narayanan (2022) train a random-forest classifier to identify the change address using as features the individual decisions of each heuristic. The source code used for both the heuristics and the model training is not publicly available, but the training dataset containing (transaction identifier, index of the change output) pairs is.

We implemented all heuristics introduced in Möser & Narayanan (2022) and selected 30,000 transactions from the publicly released dataset to train our classifier. To ensure full reproducibility, we provide our complete implementation in the same repository, together with an augmented dataset containing, for each transaction, all features required to apply the heuristics. These heuristics are summarized in Table 1 of Möser

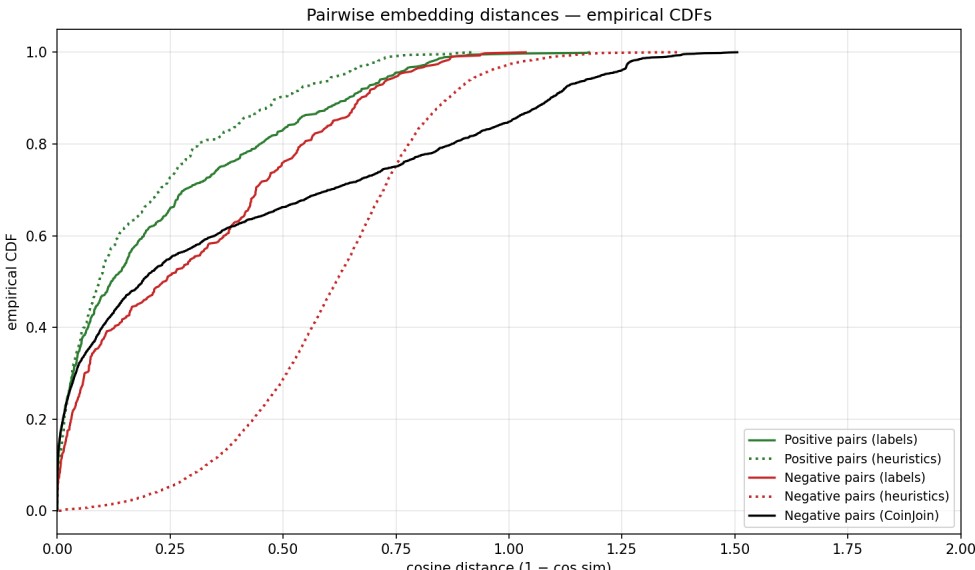

Figure 6: Empirical CDFs of cosine distances between embedding pairs with CoinJoin hard negatives.

& Narayanan (2022). The Random Forest classifier was trained using `scikit-learn`. The training dataset includes, for each transaction, the individual heuristic decisions encoded as $-1$, $0$, or $1$ (see Section 3.3 and Appendix E of Möser & Narayanan (2022)), together with additional transaction-level features described in Appendix E. We use the optimal hyperparameters reported in Appendix E of Möser & Narayanan (2022). On a 20% held-out test set, the classifier achieves an accuracy of 99.43%, consistent with the performance reported in the original study.

For experiments involving ground-truth labels, we follow the clustering procedure described in Section 4 of Möser & Narayanan (2022). Specifically, we apply the common-input heuristic and augment it with change-address predictions produced by our classifier. An address is classified as change and merged into the input cluster whenever the predicted probability exceeds 0.99, matching the conservative threshold used in Section 4 of Möser & Narayanan (2022). For the experiment involving entity labels, we construct a dataset following the same format as Table 5 in Möser & Narayanan (2022), enabling full reproducibility. In the CoinJoin experiment, the methodology achieves the same performance as the base heuristics, as it ultimately relies on the common-input heuristic, which is known to be systematically misled by CoinJoin transactions (see Sections 2 and 4 of Möser & Narayanan (2022)).

## F   Proofs of section 4

### F.1   Proof of Lemma 1.

We begin by deriving a few spectral properties of the Laplacian $L^\circ$ of the *ideal cluster graph*, in which two nodes are adjacent if and only if they belong to the same cluster. It is well known that the Laplacian $L^\circ$ of this ideal graph has 0 as an eigenvalue with multiplicity equal to the number of connected components—equivalently, the number of clusters (Von Luxburg, 2007). For each cluster $C_j$, the normalized indicator vector

$$u_{j,i}^\circ = \begin{cases} |C_j|^{-1/2}, & \text{if } i \in C_j, \\ 0, & \text{otherwise,} \end{cases}$$

is an eigenvector associated with the eigenvalue 0, and these vectors form an orthonormal basis of the corresponding eigenspace.

The spectral embedding of node $i$ in the *ideal model*, using the first $k$ eigenvectors, is its coordinate vector in this basis:

$$(e_i^\circ)_j = \begin{cases} |C_j|^{-1/2}, & \text{if } i \in C_j, \\ 0, & \text{otherwise.} \end{cases}$$

Consequently, if $i, j \in C_a$ then $e_i^\circ = e_j^\circ$; and if $i \in C_a$ and $j \in C_b$ with $a \neq b$,

$$\|e_i^\circ - e_j^\circ\|_2^2 = \frac{1}{|C_a|} + \frac{1}{|C_b|} \qquad \Rightarrow \qquad \|e_i^\circ - e_j^\circ\|_2 \geq \sqrt{\frac{2}{S_{\max}}}.$$

We now view the empirical Laplacian $L$ as a perturbation of the ideal Laplacian $L^\circ$ and invoke spectral perturbation theory. Let $U_k, U_k^\circ \in \mathbb{R}^{n \times k}$ collect the eigenvectors associated with the $k$ smallest eigenvalues of $L$ and $L^\circ$, respectively. By the Davis–Kahan–type result of Yu et al. (2015), there exists an orthogonal matrix $Q \in \mathbb{R}^{k \times k}$ such that

$$\|U_k - U_k^\circ Q\|_F \leq \frac{2\sqrt{2k}\,\|L - L^\circ\|_{\mathrm{op}}}{\lambda_{k+1}(L^\circ)},$$

where $\|\cdot\|_F$ is the Frobenius norm and $\lambda_{k+1}(L^\circ)$ denotes the $(k+1)$-th eigenvalue of $L^\circ$.

Let $e_i^s$ be the spectral embedding of node $i$ obtained from $L$. Applying the bound row-wise gives, for every node $i$,

$$\|e_i^s - e_i^\circ Q\|_2 \leq \frac{2\sqrt{2k}\,\|L - L^\circ\|_{\mathrm{op}}}{\lambda_{k+1}(L^\circ)}.$$

By the triangle inequality and the orthogonality of $Q$,

$$\|e_i^s - e_j^s\|_2 \leq \|e_i^s - e_i^\circ Q\|_2 + \|(e_i^\circ - e_j^\circ)Q\|_2 + \|e_j^\circ Q - e_j^s\|_2$$

$$\leq \frac{4\sqrt{2k}\,\|L - L^\circ\|_{\mathrm{op}}}{\lambda_{k+1}(L^\circ)} + \|e_i^\circ - e_j^\circ\|_2.$$

In particular, if $i$ and $j$ lie in the same cluster, then $e_i^\circ = e_j^\circ$ and

$$\|e_i^s - e_j^s\|_2 \leq \frac{4\sqrt{2k}\,\|L - L^\circ\|_{\mathrm{op}}}{\lambda_{k+1}(L^\circ)}.$$

A symmetric argument yields the complementary lower bound

$$\|e_i^s - e_j^s\|_2 \geq \|e_i^\circ - e_j^\circ\|_2 - \frac{4\sqrt{2k}\,\|L - L^\circ\|_{\mathrm{op}}}{\lambda_{k+1}(L^\circ)}.$$

Hence, if $i$ and $j$ belong to different clusters,

$$\|e_i^s - e_j^s\|_2 \geq \sqrt{\frac{2}{S_{\max}}} - \frac{4\sqrt{2k}\,\|L - L^\circ\|_{\mathrm{op}}}{\lambda_{k+1}(L^\circ)}.$$

Finally, for the ideal cluster graph—a disjoint union of cliques—one has $\lambda_{k+1}(L^\circ) = \frac{S_{\max}}{S_{\max}-1}$, recovering the explicit constant used earlier.

## F.2  Proof of Theorem 2.

Recall that $U_k \in \mathbb{R}^{n \times k}$ is the matrix whose columns are the $k$ orthonormal eigenvectors of $L$ associated with its $k$ smallest eigenvalues. Let $U_k^\perp$ denote the matrix whose columns form an orthonormal basis of the orthogonal complement of $\mathrm{span}(U_k)$. The block matrix

$$U := \begin{bmatrix} U_k & U_k^\perp \end{bmatrix}$$

is therefore orthogonal and provides a full orthonormal basis of $\mathbb{R}^n$.

Under our structural assumption on the GNN, for input features $X \in \mathbb{R}^{n \times d}$ and weight matrix $W \in \mathbb{R}^{d \times m}$, the linearized GNN can be written

$$H = p(L)\, XW,$$

where $p$ is a polynomial filter. Using the spectral decomposition $L = UDU^\top$, this becomes

$$H = U\, p(D)\, U^\top XW.$$

This representation naturally separates the embedding into its low-frequency and residual components,

$$H = U_k\, p(D_k)\, U_k^\top XW + U_k^\perp\, p(D_k^\perp)\, (U_k^\perp)^\top XW,$$

highlighting the projection of $H$ onto the informative subspace spanned by $U_k$ and its complement along $U_k^\perp$.

Let $P_k := U_k U_k^\top$ denote the orthogonal projector onto the eigenspace spanned by $U_k$. Then

$$(I - P_k)H \;=\; U_k^\perp\, p(D_k^\perp)\, (U_k^\perp)^\top XW,$$

so the leakage of $H$ outside $\mathrm{span}(U_k)$ is controlled by

$$\|(I - P_k)H\|_{\mathrm{op}} = \|\, p(D_k^\perp)\, (U_k^\perp)^\top XW \,\|_{\mathrm{op}}$$
$$\leq \|p(D_k^\perp)\|_{\mathrm{op}}\, \|XW\|_{\mathrm{op}}.$$

Because $D_k^\perp$ is diagonal with entries given by the eigenvalues $\lambda_{k+1}, \ldots, \lambda_n$ of $L$, the operator norm of $p(D_k^\perp)$ is simply the largest absolute value of $p(\lambda_i)$ for $i > k$. Hence

$$\|(I - P_k)H\|_{\mathrm{op}} \leq \left( \max_{i>k} |p(\lambda_i)| \right) \|XW\|_{\mathrm{op}} = \beta \,\|XW\|_{\mathrm{op}}.$$

Since for any matrix $A$ one has $\max_i \|A_{i,:}\|_2 \leq \|A\|_{\mathrm{op}}$, it follows that for each node $i$, whose embedding is the $i$-th row $h_i$ of $H$,

$$\| h_i - (P_k H)_{i,:} \|_2 \;\leq\; \|(I - P_k)H\|_{\mathrm{op}} \;\leq\; \beta \,\|XW\|_{\mathrm{op}}.$$

Let $Z := U_k^\top H \in \mathbb{R}^{k \times m}$; then $P_k H = U_k Z$, so that $(P_k H)_{i,:} = (e_i^s)^\top Z$. Therefore, for any nodes $i, j \in V$,

$$\|h_i - h_j\|_2 \leq \|h_i - (P_k H)_{i,:}\|_2 + \|(e_i^s - e_j^s)^\top Z\|_2 + \|(P_k H)_{j,:} - h_j\|_2$$
$$\leq 2\beta \,\|XW\|_2 + \|(e_i^s - e_j^s)^\top Z\|_2.$$

Since $Z = U_k^\top H = p(D_k)\, U_k^\top XW$, we obtain

$$\|Z\|_{\mathrm{op}} \;\leq\; \|p(D_k)\|_{\mathrm{op}}\, \|U_k^\top XW\|_{\mathrm{op}} \;\leq\; \left( \max_{i \leq k} |p(\lambda_i)| \right) \|XW\|_{\mathrm{op}} = \alpha \,\|XW\|_{\mathrm{op}}.$$

Consequently,

$$\|(e_i^s - e_j^s)^\top Z\|_2 \;\leq\; \|e_i^s - e_j^s\|_2\, \|Z\|_{\mathrm{op}} \;\leq\; \alpha \,\|XW\|_{\mathrm{op}}\, \|e_i^s - e_j^s\|_2.$$

Combining the two displays gives the upper bound

$$\|h_i - h_j\|_2 \;\leq\; \|XW\|_{\mathrm{op}} \left( 2\beta + \alpha \,\|e_i^s - e_j^s\|_2 \right).$$

A symmetric lower bound follows from the reverse triangle inequality:

$$\|h_i - h_j\|_2 \;\geq\; \|(e_i^s - e_j^s)^\top Z\|_2 - \|h_i - (P_k H)_{i,:}\|_2 - \|h_j - (P_k H)_{j,:}\|_2.$$

Using the fact that $\|(e_i^s - e_j^s)^\top Z\|_2 \geq \sigma_{\min}(Z)\,\|e_i^s - e_j^s\|_2$ and recalling that $Z = p(D_k)\,U_k^\top XW$, we obtain

$$\sigma_{\min}(Z) \;\geq\; \left(\min_{i \leq k} |p(\lambda_i)|\right) \sigma_{\min}(U_k^\top XW) = \gamma\,\sigma_{\min}(U_k^\top XW).$$

Hence,

$$\|h_i - h_j\|_2 \;\geq\; \gamma\,\sigma_{\min}(U_k^\top XW)\,\|e_i^s - e_j^s\|_2 - 2\,\beta\,\|XW\|_{\mathrm{op}}.$$

Using the bounds from Lemma 1 and substituting them into the inequalities above, we obtain the following estimates.

For nodes $i, j$ in the *same* cluster,

$$\|h_i - h_j\|_2 \;\leq\; \|XW\|_{\mathrm{op}}\left(2\,\beta + \frac{4\sqrt{2k}\,\alpha}{\lambda_{k+1}(L^\circ)}\|L - L^\circ\|_{\mathrm{op}}\right).$$

For nodes $i, j$ in *different* clusters,

$$\|h_i - h_j\|_2 \;\geq\; \gamma\,\sigma_{\min}(U_k^\top XW)\left(\sqrt{\frac{2}{S_{\max}}} - \frac{4\sqrt{2k}}{\lambda_{k+1}(L^\circ)}\|L - L^\circ\|_{\mathrm{op}}\right) - 2\,\beta\,\|XW\|_{\mathrm{op}}.$$

## G  Computational Complexity Analysis

We summarize here the computational costs associated with each step of our pipeline. Let $N = |V|$ denote the number of nodes, $M = |E|$ the number of edges, $d$ the embedding dimension, $k_s$ the maximum number of neighbors sampled at each GNN layer, and $k_l$ the maximum size of the Leiden pre-clusters. Table 11 reports asymptotic time and memory requirements for each stage of the method. These complexity estimates are based on the `PyTorch` implementations used for the GNN components, the `SciPy` implementation used for hierarchical agglomerative clustering, and the `leidenalg` package for the Leiden pre-clustering step.

| Step | | Time | Memory |
|---|---|---|---|
| Embeddings | Forward pass | $O(Md + Nd^2)$ | $O(Nd)$ |
| | Forward pass with sampling | $O(Nk_s^{L-1}d(k_s + d))$ | $O(Nd)$ |
| Pre-clustering | Leiden algorithm | $O(M)$ | $O(N + M)$ |
| Distance Matrix | Without pre-clustering | $O(N^2d)$ | $O(N^2)$ |
| | With pre-clustering | $O(Nk_ld)$ | $O(Nk_l)$ |
| HAClustering | Linkage vector | $O(N^2)$ | $O(N^2)$ |
| Flat Clustering | Dendrogram cut | $O(N)$ | $O(N)$ |
| | Silhouette score | $O(N^2)$ | $O(N)$ |

Table 11: Complexity Analysis.

These results highlight the main computational bottlenecks. Embedding computation scales linearly in both $N$ and $M$, especially when neighbor sampling is applied. In contrast, operations involving pairwise distances or hierarchical clustering scale quadratically in $N$, which motivates the need for pre-clustering or highly local sampling strategies. As an illustration, both Monath et al. (2021) and (Dhulipala et al., 2023) propose a scalable HAC algorithm that mitigates these quadratic costs.

