# OpenReview forum: "Refining Heuristic-Based Bitcoin Address Clustering with Graph Neural Networks"
_TMLR — Rejected by TMLR_

### Review · Reviewer_2EWs · 2026-04-19

**Summary Of Contributions:**

The paper refines heuristic-based Bitcoin address clustering via a two-stage GNN pipeline: a GCN / GraphSAGE / GAT encoder trained with a supervised InfoNCE-style contrastive loss (positives = same coarse heuristic cluster, negatives = different cluster), followed by agglomerative hierarchical clustering (HAC) in the embedding space with a silhouette-selected cut threshold λ that flags and splits "cluster collapses." Three contributions are claimed: (i) a publicly released blockchain-derived dataset of large-scale Bitcoin transaction graphs with ~500k ground-truth user clusters under disjoint block-interval splits; (ii) a contrastive methodology paired with a Davies–Kahan-style separability analysis (Lemma 1, Theorem 2) giving a perfect-cut condition for linearised GNN embeddings `H = p(L) X W`; and (iii) an empirical study (dendrogram purity, NMI, ARI, CoinJoin and entity-label metrics) showing gains over the common-input heuristic and the prior heuristic-refinement baseline.

### Strengths

- **Useful dataset release with a clean two-stage methodology.** Three train + one val + one test block-interval-disjoint graphs with ~500k labelled clusters (Appendix A) is a concrete community contribution, and Section 3.1 gives a well-motivated argument for why HAC on a contrastively trained encoder is preferable to end-to-end differentiable pooling on dynamic blockchain graphs.
- **Non-trivial theoretical contribution.** Lemma 1 gives a Davies–Kahan-style spectral-perturbation bound, and Theorem 2 transfers it to linearised GNN embeddings `H = p(L) X W`. The pairwise (rather than intra-cluster-variance) framing distinguishes the result from the classical spectral-clustering line and gives a novel connection between a contrastively trained GNN encoder and a perfect dendrogram-cut condition.
- **Thorough empirical protocol.** Table 1 sweeps symmetrisation × edge features × landmarks × architecture with 5-seed error bars; Appendix D isolates embedding dim, α, and p; Appendix D.4 verifies the low-pass assumption via Rayleigh-quotient decay. The baseline set (Louvain / Leiden, GAE, DGI, and the prior heuristic-refinement baseline) is the right one, and reproducibility hygiene (full hyperparameters, baseline re-implementation details, a reproduction of the prior baseline, complexity analysis, LLM-usage disclosure) is complete.

### Weaknesses

- **Theoretical framing is under-contextualised and not empirically verified.** Section 4 claims Theorem 2 as "to our knowledge, novel," but the broader line connecting contrastive learning to spectral clustering is not discussed: [1] provides a provable spectral-contrastive recovery guarantee and [2] makes the "contrastive learning = spectral clustering on a similarity graph" equivalence explicit. Neither subsumes Theorem 2 (a *perfect-cut* dendrogram condition for linearised GNNs), but the omission makes the theoretical contribution hard to calibrate. Separately, the RHS of Theorem 2 (σ_min(U_k^⊤ X W), β, γ, S_max, ‖L − L°‖_op) is never measured on the actual Bitcoin graph, so Section 4 currently reads as suggestive rather than predictive. The paper should cite and contrast [1, 2] at the Lemma 1 / InfoNCE introduction, and either numerically check the inequality on a sampled subgraph or re-frame Section 4 as *guiding intuition*.
- **Thin positioning against recent crypto/GNN work.** The recent GNN-based Bitcoin / crypto clustering line, including the Elliptic2 line, is cited in passing only (Section 2, "Other Methods for Address Clustering"); none is promoted to a Table 2/3 baseline. Since the central claim is that a GNN *refines* heuristic clusters, at least one of these should be an empirical baseline; if direct comparison is infeasible, the paper should state why.
- **Homophily is assumed implicitly and under-examined, which likely drives the CoinJoin gap.** Section 4 and Section 5.1 lean on "same-user addresses are close" — a homophily assumption in disguise. In Table 3 the Hybrid variant substantially underperforms the GNN-HAC variant on CoinJoin true-negative rate (e.g., 8.0 vs. 25.5 for complete linkage), and the proposed "suboptimal threshold or insufficient separation" explanation is never tested; heterophily is the natural suspect. The paper should cite [4] in the homophily discussion and explicitly address whether CoinJoin locally breaks homophily, then either run the hard-negatives variant sketched in Section 6.3.2 or at minimum provide a silhouette-threshold sensitivity plot. A broader graph-learning-theory connection to the group-level separability analysis in [5] would also strengthen the Theorem 2 discussion, since [5] studies exactly when subgraph / group embeddings become separable.
- **Scalability is announced but not stress-tested.** CPU-only training on an M3 Max is attractive, but training graphs top out around ~1M nodes and the conclusion defers scalability to future work. A sharper statement of what specifically does not yet scale (embedding vs. HAC vs. threshold selection) would be valuable, and the Appendix G complexity analysis should discuss [3] (scalable approximate HAC for trillion-edge graphs) alongside the already-cited prior scalable-HAC work. Relatedly, the random-features baseline (untrained GAT beating Louvain / Leiden in Table 1) is a well-known phenomenon that the paper does not frame as such; a brief reference to the self-supervised-MLPs-on-graphs line [6] would sharpen the interpretation of that row.


---

## References

[1] J. Z. HaoChen, C. Wei, A. Gaidon, T. Ma. *Provable Guarantees for Self-Supervised Deep Learning with Spectral Contrastive Loss.* NeurIPS 2021.

[2] Z. Tan, Y. Zhang, J. Yang, Y. Yuan. *Contrastive Learning Is Spectral Clustering on Similarity Graph.* ICLR 2024.

[3] L. Dhulipala, J. Łącki, J. Lee, V. Mirrokni. *TeraHAC: Hierarchical Agglomerative Clustering of Trillion-Edge Graphs.* SIGMOD 2023.

[4] J. Zhu, Y. Yan, L. Zhao, M. Heimann, L. Akoglu, D. Koutra. *Beyond Homophily in Graph Neural Networks: Current Limitations and Effective Designs.* NeurIPS 2020.

[5] Z. Wang, Z. Zhang, C. Zhang, Y. Ye. *Subgraph Pooling: Tackling Negative Transfer on Graphs.* IJCAI 2024.

[6] Z. Wang, Z. Zhang, C. Zhang, Y. Ye. *Training MLPs on Graphs Without Supervision (SimMLP).* WSDM 2025.

**Audience:**

Yes

**Audience Explanation:**

See summary of contributions.

**Broader Impact Concerns:**

Improved Bitcoin address clustering is dual-use: it supports legitimate forensic and AML work, but it also reduces the pseudonymity guarantees that any blockchain user — including benign users — implicitly rely on. The paper does not currently include a dedicated broader-impact paragraph. I recommend a short statement that (i) acknowledges the dual-use nature of address deanonymisation, (ii) notes that the released dataset is derived from already-public blockchain data, and (iii) briefly addresses the implications of improved refinement for privacy-seeking but legitimate users. This is not a blocker for acceptance.

**Claims And Evidence:**

Yes

**Claims Explanation:**

See summary of contributions.

**Requested Changes:**

1. **Empirically validate or re-frame Theorem 2.** Report a proxy measurement of σ_min(U_k^⊤ X W), β, γ, S_max and ‖L − L°‖_op on a representative sampled subgraph and assess whether the inequality is (approximately) satisfied. If it is not tight, rephrase Section 4 as guiding intuition rather than a strict guarantee.
2. **Add at least one recent GNN-based crypto baseline in Table 2/3.** Promote one of the GNN-based Bitcoin-clustering methods or the Elliptic2 GNN that are currently cited only in Section 2 into an empirical baseline, or explicitly justify why direct comparison is infeasible.
3. **Diagnose the CoinJoin Hybrid gap.** Either run the hard-negatives variant sketched in Section 6.3.2, or provide a silhouette-threshold sensitivity plot so the reader can distinguish tuning artefact from structural limitation.
4. **Place the theoretical contribution in context and update the literature.** Cite and briefly contrast [1] and [2] at the Lemma 1 / InfoNCE introduction; cite [3] in Appendix G alongside the already-referenced prior scalable-HAC work; cite [4] in the homophily discussion of Sections 4 and 5.1 with a sentence on CoinJoin specifically; add [5] in Section 3.1 / 5.1 when motivating that same-user addresses form connected subgraphs; add [6] in Section 6.1 when discussing the untrained-GAT / random-features baseline.

---

> ### Author Response · Authors · 2026-05-14
> **Response to Reviewer 2EWs**
>
> We thank the reviewer for the detailed and constructive feedback. We agree that several aspects of the positioning and empirical analysis can be clarified and strengthened. Below we address each point in detail.
>
>
> ### **1. Theoretical framing**
>
>
> We thank the reviewer for this important remark. We agree that the conditions of Theorem 2 are highly idealized and are not expected to hold tightly on real Bitcoin transaction graphs. In particular, the theorem assumes strong cluster separability and closeness to an ideal block-diagonal graph, whereas real blockchain graphs contain noisy interactions, exchange hubs, CoinJoin transactions, and many weakly homophilic structures. We therefore agree that Section 4 is better interpreted as providing guiding intuition for why contrastive GNN embeddings may yield separable hierarchical structures, rather than as a strict predictive guarantee for the empirical setting. We renamed the section “Theoretical guiding intuitions” accordingly.
>
> ---
>
> Directly estimating the quantities appearing in Theorem 2 on large Bitcoin transaction graphs is computationally infeasible, as it would require computing spectral quantities of very large Laplacians and access to unavailable true entity-level partitions. More importantly, the theorem fundamentally derives from assumptions on intra-cluster versus inter-cluster distances in embedding space. We therefore believe that empirical distance distributions provide a more informative and interpretable proxy than attempting to numerically verify the inequality itself.
>
> Following the reviewer’s suggestion, we therefore added a new empirical analysis based on embedding-distance distributions. Concretely, we report:
>
> * the distribution of distances between address pairs sharing the same entity label,
> * the distribution of distances between ordinary negative pairs from the entity-label dataset,
> * and the distribution of distances between CoinJoin negative pairs.
>
> The corresponding cumulative distribution functions (CDFs) will be integrated into the revised paper.
>
> ---
>
> These analyses directly measure the separability assumptions underlying the theorem and provide a practical interpretation of when hierarchical clustering succeeds or fails. They strongly suggest that the previously hypothesized explanation of a suboptimal dendrogram threshold is not the primary cause of the poor CoinJoin performance. Instead, the main issue appears to stem from insufficient separation of CoinJoin negative pairs directly in the embedding space. We believe this occurs because the original contrastive loss never exposes the model to CoinJoin-specific hard negatives during training. We therefore added a new experiment in which CoinJoin negative pairs are explicitly incorporated into the contrastive loss as additional hard negatives penalizing overly similar embeddings. To avoid any leakage from the evaluation labels, we sampled 1,000 additional CoinJoin-centered graphs used exclusively for this auxiliary training loss.
>
> The resulting model substantially improves the out-of-sample CoinJoin true-negative rates, reaching above 50% TN in five out of six evaluated GNN-based settings.. This demonstrates that the framework can naturally adapt once the relevant heterophilic patterns are incorporated into the supervision signal. The detailed results are reported in the table below. Moreover, after introducing these hard negatives, the distance distribution of CoinJoin negative pairs becomes much closer to the distribution observed for ordinary negative pairs from the entity-label dataset, further supporting the interpretation that the original CoinJoin gap primarily stemmed from insufficient embedding separation rather than from an intrinsic limitation of the framework itself.
>
> | Model      | Link. | Cut  |     TN (%) | TN (%) + CoinJoin negatives |
> | ---------- | ----- | ---- | ---------: | --------------------------: |
> | Heuristics | na    | na   |        0.0 |                         0.0 |
> | GNN-HAC    | avg.  | sil. | 25.1 ± 1.4 |                  58.0 ± 2.3 |
> | GNN-HAC    | ward  | sil. | 43.9 ± 1.5 |                  57.9 ± 1.9 |
> | GNN-HAC    | com.  | sil. | 45.8 ± 1.8 |                  63.4 ± 2.1 |
> | Hybrid     | avg.  | sil. | 14.0 ± 0.7 |                  51.5 ± 2.0 |
> | Hybrid     | ward  | sil. | 11.4 ± 0.3 |                  30.2 ± 0.5 |
> | Hybrid     | com.  | sil. | 20.5 ± 0.8 |                  53.9 ± 2.1 |
>
>
> Interestingly, these new experiments also yield substantially better true-negative rates than those initially reported in the submission. We suspect that the original model may not have fully converged for this specific application.

---

> > ### Author Response · Authors · 2026-05-14
> >
> > ### **2. Positioning with respect to recent GNN-based blockchain work**
> >
> > We thank the reviewer for this remark. We agree that the positioning with respect to recent blockchain-GNN literature should be clarified further.
> >
> > To our knowledge, the only truly comparable and reproducible prior work explicitly designed to refine heuristic-based Bitcoin address clustering is the method of Möser & Narayanan (2022), which is why we used it as the main refinement baseline in Tables 2 and 3.
> >
> > Most recent GNN-based blockchain works instead focus on different downstream tasks, such as transaction classification, AML detection, or graph-level prediction, rather than node-level address clustering refinement. As a consequence, these methods do not directly produce node embeddings suitable for hierarchical address clustering and cannot be straightforwardly integrated into our refinement pipeline.
> >
> > Regarding Elliptic2 specifically, the associated works primarily study illicit-activity detection and subgraph classification rather than heuristic-cluster refinement. Their supervision signals and evaluation protocols are therefore fundamentally different from ours. We will clarify this distinction more explicitly in the revised paper and better position our contribution relative to the broader blockchain-GNN literature.
> >
> > ### **3. Homophily assumptions and the CoinJoin gap**
> >
> > We thank the reviewer for this insightful observation. We agree that CoinJoin transactions locally violate the homophily assumptions implicitly underlying both the theoretical analysis and the contrastive training objective.
> >
> > The additional embedding-distance analysis described above strongly supports this interpretation. In the original model, CoinJoin negative pairs remain substantially closer in embedding space than ordinary negative pairs from the entity-label dataset. This suggests that the model never learned to explicitly separate such heterophilic structures because the initial contrastive supervision only relied on heuristic-derived positives and negatives.
> >
> > To address this issue, we implemented the hard-negative experiment suggested in Section 6.3.2. We augment the contrastive loss with additional CoinJoin-based negative pairs, explicitly penalizing embeddings that place CoinJoin input addresses too close together. Importantly, these additional CoinJoin graphs are sampled independently from the evaluation set to avoid any leakage.
> >
> > The resulting model substantially improves the out-of-sample CoinJoin true-negative rates, reaching more than 50% TN in all evaluated settings except one.
> >
> > ### **4. Additional references and theoretical positioning**
> >
> > We thank the reviewer for these references, which will help us better situate our theoretical contribution. We will integrate them into the revised paper in the relevant positions.
> >
> > ### **Broader impact**
> >
> > We thank the reviewer for highlighting this important point. In the revised version, we added a dedicated broader-impact paragraph explicitly acknowledging this dual-use aspect. We clarify that the released dataset is entirely derived from publicly available blockchain data and does not introduce new private information, while also discussing the broader implications of increasingly effective address-clustering techniques for privacy-preserving cryptocurrency usage.

---

### Review · Reviewer_tf97 · 2026-05-01

**Summary Of Contributions:**

This paper studies Bitcoin address clustering and proposes to refine heuristic-based clusters using GNN embeddings trained with a contrastive loss. The learned embeddings are then used with hierarchical clustering to identify possible cluster collapses and produce finer-grained partitions. The paper also releases sampled Bitcoin transaction graphs and provides theoretical discussion connecting GNN embeddings, spectral clustering, and separability.

The topic is interesting and potentially useful for blockchain analysis. However, I do not find the current evidence sufficient to support the main claims. The method is largely trained to reproduce heuristic labels and then used to correct those same heuristics, which creates a circularity concern. The evaluation with real ground-truth labels is limited, and the CoinJoin results remain weak. The theoretical results rely on strong assumptions that do not appear to be empirically validated in the Bitcoin setting. Overall, I think the paper is promising but not yet ready for acceptance.

**Additional Comments:**

I appreciate the practical motivation and the attempt to combine GNN embeddings with hierarchical clustering. However, the current submission does not yet provide enough evidence that the method reliably corrects heuristic errors rather than mostly reproducing heuristic structure. I would recommend rejection in its current form, with encouragement to resubmit after stronger ground-truth validation and clearer analysis of known heuristic failure cases.

**Audience:**

Yes

**Audience Explanation:**

The problem is relevant to graph representation learning, clustering, and blockchain analytics. The idea of using learned graph embeddings to refine heuristic clusters is also practically interesting. The released dataset may be useful for researchers working on transaction graphs or financial forensics. However, the current results are not strong enough to justify the main claims, especially regarding correction of heuristic errors.

**Broader Impact Concerns:**

1. Provide more error analysis: examples where the method correctly splits a collapsed cluster, and examples where it fails.
2. Discuss scalability more concretely. The paper notes that hierarchical clustering is a bottleneck, but the proposed solution is still not clearly scalable to full blockchain-scale data.
3. Improve writing around the claim of “ground truth.” In many places, heuristic clusters are treated as labels, but they are not true user-level ground truth.

**Claims And Evidence:**

No

**Claims Explanation:**

The empirical evidence is not fully convincing. A central issue is that the GNN is trained using heuristic-derived clusters as positive/negative supervision, but the claimed goal is to detect and correct errors in those heuristic clusters. This creates a potential circularity: it is unclear when and why a model trained to preserve heuristic consistency should reliably split incorrect heuristic merges.

The main large-scale evaluation also uses heuristic clusters as ground truth, which mainly measures agreement with the heuristic rather than correctness. The additional evaluation with labeled entities is useful, but relatively limited and based on sampled local subgraphs. The reported improvement is encouraging, but not sufficient to establish robust generalization.

The CoinJoin experiment further weakens the claim. The best GNN-HAC configuration reaches only 25.5% true-negative rate, and the hybrid refinement setting performs worse, with at most 8.0% true-negative rate. Since CoinJoin is one of the clearest cases where common-input heuristics fail, this suggests that the proposed refinement is not yet reliable for important failure modes.

The theory is mathematically interesting, but its assumptions are quite strong: separability, closeness to an ideal block-diagonal graph, and suitable low-pass behavior of the GNN. The paper does not convincingly show that these assumptions hold in the actual Bitcoin graphs. Therefore, the theoretical section currently provides limited support for the practical method.

**Requested Changes:**

1. Clarify the circularity issue. The model is trained using heuristic-derived clusters but is then expected to correct errors in those same heuristics. The paper should clearly explain under what conditions the GNN can split wrongly merged users instead of simply reinforcing the heuristic partition.
2. Strengthen the ground-truth evaluation. The current labeled-entity experiment is useful but limited. More labeled data, more diverse time periods, and larger-scale evaluation are needed to support the general claim.
3. Improve evaluation on known heuristic failure cases. The CoinJoin results are weak, especially for the hybrid refinement setting. Since CoinJoin is a key failure mode of common-input clustering, the method needs stronger validation and analysis on this case.
4. Add stronger and more direct baselines. The current baselines are helpful, but the paper should also compare against metric-learning or pairwise merge-classification baselines trained with the same heuristic supervision.

---

> ### Author Response · Authors · 2026-05-14
> **Response to Reviewer tf97**
>
> We thank the reviewer for the detailed and constructive feedback. We agree that the distinction between learning heuristic-consistent embeddings and correcting heuristic failures should be stated more carefully, and we will revise the paper accordingly.
>
> ### **1. Clarifying the circularity issue**
>
> We agree that this point deserves a clearer explanation. Our objective is not to exactly reproduce the heuristic partition, but rather to use heuristics as a supervision signal alongside the graph structure for representation learning. The heuristics are only used during training to define positive and negative pairs in the contrastive loss. Importantly, these heuristics are widely used in the Bitcoin literature because they are known to be effective in the majority of cases, which is also reflected in the relatively strong baseline scores reported in Table 2.
>
> The key point is that heuristics provide only binary decisions: either two addresses belong to the same cluster or they do not. In contrast, the learned embeddings encode a richer notion of similarity based on node attributes, local transaction structure, and multi-hop graph topology. Two addresses that are grouped together by a heuristic may still end up far apart in embedding space if their transactional context differs substantially. Hierarchical clustering on these embeddings can therefore reveal suspicious merges that are not explicit in the original heuristic partition.
>
> We will clarify this intuition in the revised paper and emphasize that the large-scale experiments of Table 1 are not intended to demonstrate true entity recovery, but rather to show that the GNN successfully learns meaningful heuristic-consistent representations on unseen graphs sampled from disjoint block intervals. The actual evidence for heuristic refinement comes from Section 6.3, where evaluation relies on independent labels rather than heuristic clusters.
>
> We will also clarify throughout the paper that the entity-label experiments do not rely on heuristic clustering labels in any way.
>
> ### **2. Strengthening the ground-truth evaluation**
>
> We agree that stronger statistical framing is useful. In the revised version, we will report the mean and standard deviation over multiple random seeds and multiple sampling realizations for the entity-label experiments.
>
> To the best of our knowledge, we already use essentially all publicly available Bitcoin entity-label datasets that do not themselves rely on heuristic clustering. In particular, the dataset of Schnoering & Vazirgiannis (2025) aggregates several independent labeling sources.
>
> Regarding the use of local sampled subgraphs, we note that this is standard practice in the Bitcoin clustering and entity-prediction literature [1, 2]. The Bitcoin transaction graph is extremely sparse and highly dynamic, with a very large fraction of addresses appearing only once. As a result, many downstream analyses naturally focus on local transactional neighborhoods around transactions or addresses of interest. In our setting, local subgraphs are also important for computational tractability and are explicitly motivated in Appendix A.1.2.
>
> [1] Weber, M., Domeniconi, G., Chen, J., Weidele, D. K. I., Bellei, C., Robinson, T., & Leiserson, C. E. (1908). Anti-money laundering in bitcoin: Experimenting with graph convolutional networks for financial forensics.
>
>
> [2] Bellei, C., Xu, M., Phillips, R., Robinson, T., Weber, M., Kaler, & Chen, J. (2024). The shape of money laundering: Subgraph representation learning on the blockchain with the elliptic2 dataset.

---

> ### Author Response · Authors · 2026-05-14
>
> ### **3. Improving evaluation on heuristic failure cases**
>
> We agree that the original CoinJoin results were not sufficiently strong, especially in the hybrid refinement setting. Following the suggestion of another reviewer, we performed an additional experiment in which CoinJoin input pairs are explicitly incorporated as hard negatives during contrastive training.
>
> Concretely, we sample additional CoinJoin transactions during training and add a penalty term encouraging embeddings of CoinJoin co-input addresses to remain separated. This substantially improves out-of-sample performance on the CoinJoin benchmark, with true-negative rates now exceeding 50% across all evaluated settings except one (see below).
>
> | Model      | Link. | Cut  |     TN (%) | TN (%) + CoinJoin negatives |
> | ---------- | ----- | ---- | ---------: | --------------------------: |
> | Heuristics | na    | na   |        0.0 |                         0.0 |
> | GNN-HAC    | avg.  | sil. | 25.1 ± 1.4 |                  58.0 ± 2.3 |
> | GNN-HAC    | ward  | sil. | 43.9 ± 1.5 |                  57.9 ± 1.9 |
> | GNN-HAC    | com.  | sil. | 45.8 ± 1.8 |                  63.4 ± 2.1 |
> | Hybrid     | avg.  | sil. | 14.0 ± 0.7 |                  51.5 ± 2.0 |
> | Hybrid     | ward  | sil. | 11.4 ± 0.3 |                  30.2 ± 0.5 |
> | Hybrid     | com.  | sil. | 20.5 ± 0.8 |                  53.9 ± 2.1 |
>
> These additional experiments support the idea that the framework can naturally adapt when explicit negative supervision is provided for known heuristic failure modes. We will include these new experiments and discuss them in detail in the revised manuscript.
>
> Interestingly, these new experiments also yield substantially better true-negative rates than those initially reported in the submission. We suspect that the original model may not have fully converged for this specific application, or that one experimental parameter may have been suboptimally specified in the initial CoinJoin experiments.
>
>
> ### **4. Baselines and metric learning**
>
> We thank the reviewer for this remark.
>
>
> To our knowledge, the only truly comparable and reproducible prior work explicitly designed to refine heuristic-based Bitcoin address clustering is the method of Möser & Narayanan (2022), which is why we used it as the main refinement baseline in Tables 2.
>
>
> We agree that the positioning with respect to recent blockchain-GNN literature should be clarified further. Most recent GNN-based blockchain works instead focus on different downstream tasks such as transaction classification, AML detection, or graph-level prediction, rather than node-level address clustering refinement. As a consequence, these methods do not directly produce node embeddings suitable for hierarchical address clustering and cannot be straightforwardly integrated into our refinement pipeline.
>
> Regarding the suggested baselines, we understand it as training (for instance) a GNN-based edge predictor on the same heuristic supervision to compare both approaches, predicting an edge as "both addresses belong to the same entity". The first phase would be similar to our base approach as it recovers the embeddings, then replacing the hierarchical clustering by a supervised model to predict pairwise similarity. The main difference would be using cross-entropy rather than contrastive loss, which should align. In that case, would the reviewer be satisfied with training a supervised pairwise MLP on the obtained embeddings to compare this downstream part to the hierarchical clustering? We would appreciate the the reviewer's clarification on what exactly they expect.
>
> ### **Theoretical assumptions and Terminology**
>
> We agree that the assumptions are idealized and should not be interpreted as guarantees that exactly hold on the Bitcoin graph. Following the reviewer’s suggestion, we will therefore reframe Section 4 more explicitly as a guiding theoretical intuition connecting homophily, spectral embeddings, and low-pass GNN behavior, rather than as a direct predictive model of the empirical setting. We will also clarify the role of the empirical checks in Appendix D and soften the wording around “ground truth” throughout the paper.
>
> We agree that the wording around “ground truth” should be clarified. In the revised manuscript, we will reserve the term “ground truth” for independently labeled datasets and instead refer to heuristic-generated partitions as “heuristic labels” or “training labels”.

---

### Review · Reviewer_utFf · 2026-05-08

**Summary Of Contributions:**

This paper addresses Bitcoin address clustering, where the goal is to infer which pseudonymous addresses are controlled by the same user or entity. The authors argue that existing heuristic-based methods produce flat clusters that are difficult to inspect and can incorrectly merge unrelated users, producing cluster collapse.

The proposed method learns address embeddings with a contrastive GNN objective that encourages consistency with heuristic clusters. These embeddings are then used for hierarchical agglomerative clustering, producing dendrograms within heuristic clusters and allowing high-distance merges to be flagged or split. The paper also releases Bitcoin transaction graph datasets, provides theoretical separability conditions relating graph homophily, spectral embeddings, and GNN embeddings, and evaluates the method against several baselines, entity labels, and CoinJoin labels.

**Additional Comments:**

This is a promising paper on an important and technically interesting problem. I especially liked the idea of using learned embeddings to expose hierarchical structure inside otherwise flat heuristic clusters, and the empirical results against entity labels suggest that the method can reduce some false merges.

My main suggestion is for the paper to more carefully separate heuristic-consistency results from evidence of true heuristic refinement. The independent-label experiments are the right direction, and with clearer framing and additional uncertainty analysis, the paper could make a useful contribution to blockchain graph learning and address-clustering methodology.

**Audience:**

Yes

**Audience Explanation:**

Bitcoin address clustering is important for a wide range of applications in forensics. The idea of turning flat heuristic clusters into dendrograms is practically useful, because large heuristic clusters are difficult to inspect and a hierarchical representation can help analysts identify suspicious merges at multiple resolutions. The released dataset and code also make the paper potentially valuable to researchers working on graph representation learning for blockchain data.

**Broader Impact Concerns:**

Yes, there is a broader-impact concern for privacy.

Bitcoin address clustering can help forensic and compliance applications, but it can also reduce the practical pseudonymity of users by improving entity-level attribution. A short discussion of this dual-use aspect and appropriate safeguards would strengthen the broader-impact statement.

**Claims And Evidence:**

Yes

**Claims Explanation:**

While the overall paper is clear and the results are promising, I think the evidence for heuristic-error correction would benefit from clearer framing.

The paper provides useful evidence that graph topology and node features contain signal for learning representations consistent with heuristic-based Bitcoin address clusters. Table 1 shows that learned GNN embeddings, especially symmetrized GAT variants, improve over Louvain, Leiden, random GAT, GAE, and DGI baselines when evaluated against heuristic clustering. The ablations are also helpful and generally support the authors' implementation choices. I also appreciated the theoretical development, which gives a clean conceptual account of when GNN embeddings should preserve cluster separability through their connection to spectral embeddings and low-pass filtering.

That said, the main limitation in the evidence is that support for refining erroneous heuristic clusters is less direct and somewhat more mixed than support for learning heuristic-consistent embeddings. A substantial part of the main evaluation is based on heuristic labels: the model is trained to make embeddings consistent with heuristic clusters, and Table 1 evaluates against those clusters. This is useful for measuring heuristic consistency, but the distinction matters because the cited Schnoering and Vazirgiannis dataset paper treats heuristic clusters as likely entities while also noting failure cases such as CoinJoin. The authors do evaluate CoinJoin and entity-label settings in Section 6.3, which is helpful, but the evidence for correction of heuristic mistakes is more mixed.

The independent-label experiments in Section 6.3 are promising, especially the entity-label result where the hybrid average-linkage/silhouette configuration improves macro-F1 from 59.2% to 69.7% and balanced accuracy from 59.1% to 70.3%. I would find this evidence stronger with confidence intervals, statistical tests, or a sensitivity analysis for the sampled labeled pairs. As mentioned earlier, even though the absolute TN rates remain modest, the CoinJoin experiment points in a useful direction.

Overall, I find the idea compelling, but I would like the claims about correction of heuristic failures to be stated more explicitly and carefully. I believe this is achievable in a focused revision and would make the paper stronger.

**Requested Changes:**

The suggestions below are intended to make the strongest parts of the paper clearer and to better align the claims with the evidence.

### Content

- [C1] The paper should more clearly distinguish evaluation against heuristic clusters from evaluation against independent ground truth. Table 1 is useful, but it primarily shows that the learned embeddings recover heuristic structure. Claims about true entity recovery should rely on Section 6.3.

- [C2] The entity-label experiment in Section 6.3 is promising, and would be stronger with uncertainty estimates. Confidence intervals, statistical tests, or a sensitivity analysis for the pair-sampling cap would make the result easier to assess.

- [C3] The CoinJoin experiment is helpful, but a short discussion of what failure modes remain would make the result easier to interpret.

- [C4] Section 5.2 describes selecting a cut threshold by global silhouette score, while Appendix C.2 describes local thresholds aggregated by size-weighted averaging. Since the refinement depends on dendrogram cuts, the exact procedure and its sensitivity should be clearer.

- [C5] Theoretical results and proofs are interesting and contribute well to the overall discussion. As a small clarification, it would help to explain how readers should interpret the empirical checks in Appendix D relative to the idealized assumptions in the theoretical results.


### Editorial

- [E1] The phrase 'ground truth' for supervised learning is potentially misleading when referring to heuristic-generated clusters. "Training labels" or "heuristic labels" would be clearer, though I would be interested in the authors' perspective on this terminology.

- [E2] Table 3's caption appears inconsistent with the table contents. The caption mentions average linkage combined with three dendrogram cut methods, but the table reports average, Ward, and complete linkage with silhouette cuts.

- [E3] Table 1 lists "DIG", which appears to be a typo for DGI.

- [E4] Some claims could be softened slightly. For example, statements that the method "corrects" collapses should acknowledge that the split is embedding- and threshold-dependent and only partially validated against independent labels.

- [E5] The current supplementary README appears to contain some inconsistent paths. For the final version, please ensure these are fixed so the data location, splits, preprocessing steps, and commands are easy to verify.

Addressing C1-C4 would make the primary refinement claims better supported. C5 and the editorial comments are smaller clarifications that would fix some minor issues, and strenghten the final paper.

---

> ### Author Response · Authors · 2026-05-14
> **Response to Reviewer utFf**
>
> We thank the reviewer for the careful reading and constructive feedback. We especially appreciate the distinction drawn between heuristic-consistency and independent evidence of entity recovery, and we agree that clarifying this point substantially strengthens the paper.
>
> ### **[C1] Distinguishing heuristic-consistency from independent ground truth**
>
> We agree with the reviewer that Table 1 primarily evaluates the extent to which the learned embeddings recover and structure the heuristic partition, rather than directly measuring true entity recovery. In the revised version, we clarify this distinction throughout Sections 5 and 6.
>
> More specifically, we now explicitly present Table 1 as evaluating *heuristic-consistency* and representation quality with respect to the training labels, while reserving claims regarding recovery of true user structure to the independent-label experiments of Section 6.3. We also soften several formulations suggesting that the method universally “corrects” heuristic failures.
>
> ### **[C2] Uncertainty estimates for the entity-label experiment**
>
> We agree that the entity-label experiment would benefit from uncertainty estimates and sensitivity analyses.
>
> In the revised version, we now report the mean and standard deviation over 5 random seeds for the experiments of Section 6.3. This allows us to evaluate the robustness of the reported improvements with respect to both initialization randomness and the pair-sampling procedure, and to better quantify the statistical significance of the gains observed in Table 2.
>
> ### **[C3] Discussion of remaining CoinJoin failure modes**
>
> We thank the reviewer for this suggestion. As also requested by another reviewer, we extended the CoinJoin experiments by explicitly incorporating CoinJoin-derived hard negatives into the contrastive training procedure.
>
> More precisely, we constructed an additional dataset of CoinJoin transactions disjoint from those used in Section 6.3, and used input addresses from these transactions as explicit negative pairs during training. Preliminary results show substantially improved robustness on CoinJoin-related failure cases, particularly in the hybrid refinement setting (see below).
>
>
> | Model      | Link. | Cut  |     TN (%) | TN (%) + CoinJoin negatives |
> | ---------- | ----- | ---- | ---------: | --------------------------: |
> | Heuristics | na    | na   |        0.0 |                         0.0 |
> | GNN-HAC    | avg.  | sil. | 25.1 ± 1.4 |                  58.0 ± 2.3 |
> | GNN-HAC    | ward  | sil. | 43.9 ± 1.5 |                  57.9 ± 1.9 |
> | GNN-HAC    | com.  | sil. | 45.8 ± 1.8 |                  63.4 ± 2.1 |
> | Hybrid     | avg.  | sil. | 14.0 ± 0.7 |                  51.5 ± 2.0 |
> | Hybrid     | ward  | sil. | 11.4 ± 0.3 |                  30.2 ± 0.5 |
> | Hybrid     | com.  | sil. | 20.5 ± 0.8 |                  53.9 ± 2.1 |
>
>
> Interestingly, these new experiments also yield substantially better false-negative rates than those initially reported in the submission. We suspect that the original model may not have fully converged for this specific application, or that one experimental parameter may have been suboptimally specified in the initial CoinJoin experiments.
>
> ---
>
> We additionally expanded the discussion of remaining failure modes. In particular, we now clarify that some CoinJoin transactions still induce embeddings that remain insufficiently separated under hierarchical clustering, either because of strong local structural similarity.  Under the standard training procedure, input addresses involved in the same CoinJoin transaction were mapped to embeddings that were often much closer to each other than ordinary negative pairs from the entity-label dataset. As a result, the HAC step had little geometric signal on which to separate them reliably. Adding CoinJoin input pairs as explicit hard negatives during training directly addresses this issue: it pushes these heterophilic pairs apart in embedding space and leads to the improved CoinJoin true-negative rates reported above. In the revised paper, we will add a dedicated analysis of embedding-distance distributions for several pair types, including entity-label positives, entity-label negatives, and CoinJoin negatives. Beyond diagnosing the CoinJoin failure mode, this analysis provides an empirical proxy for the separation assumptions underlying the theoretical intuition.

---

> ### Author Response · Authors · 2026-05-14
>
> ### **[C4] Clarification of the dendrogram cut procedure**
>
> We agree that the original description of the threshold-selection procedure was ambiguous.
>
> In the revised manuscript, we clarify that the cut parameter $\lambda$ is not obtained from a single global silhouette optimization over the entire graph. Instead, for each Leiden community, we independently construct a hierarchical clustering over the embeddings and search over a geometric grid of candidate thresholds. We select the threshold maximizing the silhouette coefficient *within that community*, under the constraint that the induced number of clusters lies between 10 and \(n-1\) in order to avoid degenerate cuts.
>
> The resulting local thresholds are then aggregated through a size-weighted average over admissible communities. Communities that are too small or that admit no valid cut are excluded from this aggregation. The resulting aggregated value is finally used as a single global distance threshold during refinement.
>
>
>
> ### **[C5] Interpreting Appendix D relative to the theoretical assumptions**
>
> We thank the reviewer for highlighting this point.
>
> The theoretical results of Section 4 rely on idealized assumptions, including approximate homophily, spectral separability, and low-pass behavior of the GNN. The empirical analyses of Appendix D are not intended as formal verification of the assumptions of Theorem 2, but rather as qualitative evidence that the learned embeddings exhibit behaviors compatible with the theoretical framework.
>
> In the revised version, we explicitly clarify this interpretation. In particular, Appendix D is now framed as providing empirical indicators supporting the plausibility of the assumptions on real Bitcoin graphs, rather than as demonstrating that the theoretical inequalities hold exactly in practice.
>
> ### **[E1] Terminology around “ground truth”**
>
> We agree that the terminology could be misleading in some parts of the paper. In the revised version, we replace several occurrences of “ground truth” by “heuristic labels” or “training labels” when referring to heuristic-generated clusters, reserving “ground truth” for independently labeled datasets.
>
> ### **[E2] Table 3 caption inconsistency**
>
> We thank the reviewer for spotting this inconsistency. We corrected the caption of Table 3 to accurately reflect the linkage criteria and cut methods reported in the table.
>
> ### **[E3] Typo “DIG” instead of “DGI”**
>
> We corrected the typo in Table 1.
>
> ### **[E4] Overly strong claims regarding heuristic correction**
>
> We agree that some formulations were overly strong. In the revised manuscript, we soften several claims regarding heuristic “correction” and explicitly acknowledge that the refinement procedure is embedding- and threshold-dependent, with only partial validation against independent labels.
>
> ### **[E5] Supplementary README inconsistencies**
>
> We thank the reviewer for pointing this out. We revised the supplementary README and repository paths to ensure consistency and reproducibility, including clearer instructions regarding data locations, preprocessing, dataset splits, and experiment execution.
>
> ### **Broader impact**
>
> We thank the reviewer for raising this important point. In the revised manuscript, we therefore added a dedicated broader-impact discussion explicitly acknowledging these privacy implications. We also clarify that all experiments rely exclusively on publicly available blockchain data and do not involve deanonymization through external private information. Finally, we emphasize that the purpose of this work is to study the limitations and failure modes of existing heuristic clustering methods, including privacy-related weaknesses such as CoinJoin resistance, rather than to provide a production-ready deanonymization system.

---

### Decision · Action_Editor_4S1o · 2026-06-19

**Recommendation:** Reject

**Audience:**

Yes

**Audience Explanation:**

The research direction is interesting but the experimental section is a bit premature

**Claims And Evidence:**

No

**Claims Explanation:**

The paper present some interesting ideas and results although the experimental sections should be expanded before acceptance.

In particular, the point raised by reviewer tf97 have only been partially addressed in the rebuttal phase and the experiment are run only on small networks with less than 1M of nodes making the results less appealing.

Overall, there is a need of more detailed experimental analysis on larger graphs and with additional baselines to fully evaluate the algorithm.

**Resubmission Of Major Revision:**

The authors may consider submitting a major revision at a later time.